

**Contributions of different sources to nitrous acid (HONO) at the SORPES**
**station in eastern China: results from one-year continuous observation**
Yuliang Liu[1,2], Wei Nie[1,2*], Zheng Xu[1,2], Tianyi Wang[1,2], Ruoxian Wang[1,2],
Yuanyuan Li[1,2], Lei Wang[1,2], Xuguang Chi[1,2], and Aijun Ding[1,2]
[1] Joint International Research Laboratory of Atmospheric and Earth System Sciences, School
of Atmospheric Sciences, Nanjing University, Nanjing, Jiangsu Province, China
[2] Collaborative Innovation Center of Climate Change, Jiangsu Province, China
**Abstract**
Nitrous acid (HONO), a reservoir of the hydroxyl radical (OH), has been
long-standing recognized to be of significant importance to atmospheric chemistry,
but its sources are still debate. In this study, we conducted continuous measurement of
HONO from November 2017 to November 2018 at SORPES station in Nanjing of
eastern China. The yearly average mixing ratio of observed HONO was $0.69 \pm 0.58$
ppb, showing a larger contribution to OH relative to ozone with a mean OH
production rate of $0.90 \pm 0.27$ ppb/h. To estimate the effect of combustion emissions
of HONO, the emitted ratios of HONO and $NO_x$ were derived from 55 fresh plumes
($NO/NO_x > 0.85$), with a mean value of 0.79%. The well-defined seasonal and diurnal
patterns with clear wintertime and early morning concentration peaks of both HONO
and $NO_x$ indicate that $NO_x$ is the critical precursor of HONO. During the nighttime,
the chemistry of HONO was found to depend on RH, and heterogeneous reaction of
$NO_2$ on aerosol surface was presumably responsible for HONO production. The
average nighttime $NO_2$-to-HONO conversion frequency ($C_{HONO}$) was determined to
be $0.0055 \pm 0.0032$ $h^{-1}$ from 137 HONO formation cases. The missing source of
HONO around noontime seemed to be photo-induced with an average $P_{unknown}$ of 1.13
$\pm 0.95$ ppb $h^{-1}$, based on a semiquantitative HONO budget analysis. An
over-determined system of equations was applied to obtain the monthly variations in
nocturnal HONO sources. Except for burning-emitted HONO (approximately 23%
of total measured HONO), the contribution of heterogeneous formation on ground
surfaces was an approximately constant proportion of 36% throughout the year. The



soil emission revealed clear seasonal variation, and contributed up to 40% of observed
HONO in July and August. A higher propensity for generating HONO on aerosol
surface occurred in heavily polluted period (about 40% of HONO in January). Our
results highlight ever-changing contributions of HONO sources, and encourage more
long-term observations to evaluate the contribution from varied sources.
**1. Introduction**
Nitrous acid (HONO) is a vital constituent of nitrogen cycle in the atmosphere, first
observed in the field by Perner and Platt (1979). The concentrations of HONO varied
from dozens of ppt in remote regions (Villena et al., 2011b;Meusel et al., 2016) to
several ppb in polluted urban regions (Yu et al., 2009;Tong et al., 2015). The
photolysis of HONO (R1) has been long standing as a momentous source of the
hydroxyl radical (OH) especially during the early morning when other OH sources are
minor (Platt et al., 1980;Alicke, 2002, 2003). Even during the daytime, recent studies
have recognized the photolysis of HONO as a potentially stronger contributor to
daytime OH radical than that of $O_3$ (Kleffmann, 2005;Elshorbany et al., 2009;Li et al.,
2018). Meanwhile, HONO has been found to affect adversely human heath (Jarvis et
al., 2005;Sleiman et al., 2010).

Although the significance of HONO has been given much weight, the sources of
ambient HONO are complicated and have been debated for decades. HONO can be
emitted from combustion, including vehicle exhaust, industrial exhaust and biomass
burning (Table 1). Tunnel experiments with tests for different engine types have
determined an emission ratio of $HONO/NO_x$ for traffic source, ranged in 0.3-0.8%
(Kirchstetter et al., 1996;Kurtenbach et al., 2001). The release from soil nitrite
through acidification reaction and partitioning is considered to be another primary
source of atmospheric HONO (Su et al., 2011). Soil nitrite could come from
biological nitrification and denitrification processes (Canfield et al., 2010;Oswald et
al., 2013), or be enriched via reactive uptake of HONO from the atmosphere
(VandenBoer et al., 2014a;VandenBoer et al., 2014b). In addition to direct emissions,



the vast majority of HONO is produced chemically. The recombination of NO and
OH (R3) is the main homogeneous reaction for supplying HONO (Pagsberg et al.,
1997;Atkinson, 2000), whose contribution may be significant under conditions of
sufficient reactants during daytime. During the nighttime, with low OH concentrations,
other larger sources, heterogeneous reactions of $NO_2$ on various surfaces, are required
to explain elevated mixing levels of HONO. Laboratory studies indicate that $NO_2$ can
be converted to HONO on humid surfaces (R4), being first order in $NO_2$ and
depending on various parameters including the gas phase $NO_2$ concentration, the
surface water content, and the surface area density (Kleffmann et al.,
1998;Finlayson-Pitts et al., 2003). Besides, heterogeneous reduction of $NO_2$ with
surface organics (R5) is proposed to be another effective pathway to generate HONO
(Ammann et al., 1998;Ammann et al., 2005;Aubin and Abbatt, 2007), observed in
freshly emitted plumes with high concentrations of $NO_x$ and BC (Xu et al., 2015).
Notably this reaction rate is drastically reduced after the first few seconds due to
consumption of the reactive surfaces (Kalberer et al., 1999;Kleffmann et al., 1999),
but this reaction could be strongly enhanced by light on photo-activated surface
(George et al., 2005;Stemmler et al., 2006;Stemmler et al., 2007). During the daytime,
heterogeneous HONO formation from the photolysis of adsorbed nitric acid ($HNO_3$)
and particulate nitrate ($NO_3^-$) at UV wavelengths has been found in experiments and
observations (Zhou et al., 2003;Zhou et al., 2011;Ye et al., 2016;Ye et al., 2017).
Heterogeneous processes are typically considered as the primary sources of HONO in
many regions yet are the most poorly understood. For $NO_2$ conversion to HONO on
surfaces (R4, R5), the uptake coefficients of $NO_2$ derived from different experiments
vary from $10^{-9}$ to $10^{-2}$ (Ammann et al., 1998;Kirchner et al., 2000;Underwood et al.,
2001;Aubin and Abbatt, 2007;Zhou et al., 2015). The key step to determine the
uptake of $NO_2$ or the reaction rate is still ill-defined, and we are also not certain if and
how the ambient natural surfaces can be reactivated by radiation. Furthermore, it has
become a main concern to compare the contributions of ground and aerosol surfaces
to HONO formation. It is so far, not well explained for the observed HONO,
especially during daytime. Large unknown sources of HONO were identified by many



studies (Su et al., 2008b;Sörgel et al., 2011;Michoud et al., 2014;Lee et al., 2016).

Benefitting from more and more studies, particularly the observations under different
environment (Lammel and Cape, 1996;Li et al., 2012), understanding of HONO
chemistry in the atmosphere has been greatly improved during the last decade.
However, most HONO observations were short-term campaigns with studies ranging
from several weeks to several months. For example, Reisinger (2000) found a linear
correlation between the $HONO/NO_2$ ratio and aerosol surface density in the polluted
winter atmosphere, and Nie et al. (2015) showed the influence of biomass burning
plumes on HONO chemistry, according to observed data during late April–June 2012,
while Wong et al. (2011) believed that $NO_2$ to HONO conversion on the ground was
the dominant source of HONO by analyzing vertical profiles from 15 August to 20
September in 2006. Moreover, a theory that HONO from soil emission explained the
strength and diurnal variations of the missing source has been presented by Su et al.
(2011) based on data measured from 23 to 30 October 2004. In case the HONO
sources possibly exhibit temporal variability, especially seasonal differences, it is
challenging to draw a full picture on the basis of these short-term observations. More
than a year of continuous observation is needed, yet rather limited.

The Yangtze River Delta (YRD) is one of the most developed regions in eastern
China. Rapid urbanization and industrialization have induced severe air pollution over
the last three decades, particularly high concentrations of reactive nitrogen (Richter et
al., 2005;Rohde and Muller, 2015), including HONO (Wang et al., 2013;Nie et al.,
2015). In this study, we conducted continuous HONO observations at the SORPES
station (Station for Observation Regional Processes and the Earth System), located in
the western part of the YRD, a place that can be influenced by air masses from
different source regions of anthropogenic emissions, biomass burning, dust and
biogenic emissions (Ding et al., 2013;Ding et al., 2016). Our observation was
conducted continuously from November 2017 to November 2018 and showed
well-defined diurnal patterns and obvious season variations of HONO concentrations



at relatively high levels. We discussed the potential mechanism of HONO production
based upon semiquantitative analysis and correlation studies, and paying special
attention to changes in major sources of HONO during different seasons.

**2. Methodology**

*2.1. Study site and instrumentation*

Continuously observations was conducted at the SORPES station at the Xianlin
Campus of Nanjing University (118°57′E, 32°07′N), located in the northeast suburb
of Nanjing, China, from November 2017 to November 2018. The easterly prevailing
wind and synoptic condition makes it a representative background site of Nanjing and
a regional, downwind site of the city cluster in YRD region. Detailed descriptions for
the station can be found in previous studies (Ding et al., 2013;Ding et al., 2016).

HONO was measured with a commercial long path absorption photometer instrument
(QUMA, Model LOPAP-03). A brief description of this instrument is provided as
follows. The ambient air was sampled in two similar temperature controlled stripping
coils in series using a mixture reagent of 100 g sulfanilamide and 1 L HCl (37%
volume fraction) in 9 L pure water. In the first stripping coil, almost all of the HONO
and a fraction of interfering substances were absorbed into solution, and the
remaining HONO and the most of the interfering species were absorbed in the second
stripping     coil.     After     adding     a     reagent     of     1.6     g
N-naphtylethylendiamine-dihydrochloride in 9 L pure water to both coils, colored azo
dyewas formed in the solutions from 2 stripping coils, which were then separately
detected via long path absorption in special Teflon tubing. The interference free
HONO signal was the difference between the signals in the two channels. Further
details can be found in (Heland et al., 2001;Kleffmann et al., 2006). To correct for the
small drifts in instrument's baseline, compressed air was sampled every 12 h (flow
rate: 1 L/min) to make zero measurement. A span check was made using 0.04 mg/m$^3$
nitrite (NO$_2^-$) solution each weeks with a flow rate of 0.28 ml/min. The time



resolution, detection limit, accuracy of the measurement was 10 min, 5 pptv, and 10%,
respectively.

The NO and $NO_2$ levels were measured using a chemiluminescence instrument (TEI,
model 42i) coupled with a highly selective photolytic converter (Droplet
Measurement Technologies, model BLC), and the analyzer had a detection limit of 50
pptv for an integration time of 5 min, with precision of 4% and an uncertainty of 10%
(Xu et al., 2013). $O_3$ and CO were measured continuously using Thermo-Fisher
Scientific TEI 49i and TEI 48i. The fine particle mass concentration ($PM_{2.5}$) was
continuously measured with a combined technique of light scattering photometry and
beta radiation attenuation (Thermo Scientific SHARP Monitor Model 5030). Water
soluble aerosol ions ($NO_3^-$, $SO_4^{2-}$, $NH_4^+$ etc.) and ammonia ($NH_3$) were measured by a
Monitor for Aerosols and Gases in ambient Air (designed and manufactured by
Applikon Analytical B.V., the Netherlands) with a $PM_{2.5}$ cyclone inlet, in a time
resolution of1 hr. The size distribution of submicron particles (6-820 nm) is measured
with a DMPS (differential mobility particle sizer) constructed at the University of
Helsinki in Finland. Meteorological measurements including relative humidity (RH),
wind speed, wind direction, and air temperature were recorded by Automatic Weather
Station (CAMPEEL co., AG1000). UVB total radiation was measured by UVB
radiometer (UVS-B-T UV Radiometer, KIPP &ZONEN).

*2.2. TUV model and OH estimate*

The Tropospheric Ultraviolet and Visible (TUV) Radiation Model (http://www.acd.
ucar.edu/TUV) was adopted to compute the photolysis frequencies, which is
most probably accurate in clean and cloudless days. The pivotal parameters of this
model were inputted as follows: the ozone density was measured by Total Ozone
Mapping Spectrometer (http://toms.gsfc.nasa.gov/teacher/ozoneoverhead.html); the
typical   single scattering albedo (SSA) and Ångström exponent (Alpha) were 0.93
and 1.04 (Shen et al., 2018); the mean value of optical depth (AOD) at 550 nm was
0.640,   derived following an empirical relationship with $PM_{2.5}$ in Nanjing (Shao et



al., 2017). To reduce the error of model, we used observed UVB to correct simulated
results ($J_{mod}$) by Eq. (1). The daytime OH concentration was calculated by applying
the linear fitting formula (Eq. 2) that obtained from correlations of measured OH
concentrations with simultaneously observed $J(O^1D)$, suggested by Rohrer and
Berresheim (2006). The calculated OH concentrations around noon were in the range
of $0.15\text{-}1.17 \times 10^7$ $cm^{-3}$, comparable to observations in Chinese urban atmospheres (Lu
et al., 2012;Lu et al., 2013).

$$J = \frac{UVB_{obs}}{UVB_{mod}} J_{mod} \tag{1}$$

$$[OH] = a \times (J(O^1D)/10^{-5}\,s^{-1})^b + c$$
$$(a = 2.4 \times 10^6\,cm^{-3}, b = 1, c = 0.13 \times 10^6\,cm^{-3}) \tag{2}$$


**3. Results**

*3.1. Observation overview*

We carried out continuous measurements for HONO at SORPES station in the
northeast suburb of Nanjing from November 2017 to November 2018 with a mean
measured ambient HONO mixing level of 0.69 ± 0.58 ppb, within the range of those
in or in the vicinity of mega cities (Table 2). Fig. 1 shows the seasonal pattern of
HONO and related parameters. The highest concentration of HONO was found in
winter (1.04 ± 0.75 ppb), followed by spring (0.68 ± 0.48 ppb), autumn (0.66 ± 0.53
ppb) and summer (0.45 ± 0.37 ppb). Such seasonal variations in Nanjing are aligned
with that in Beijing (Hendrick et al., 2014), and are somewhat similar to those in Jinan
(Li et al., 2018), where the highest levels occurred in winter and the lowest levels
occurred in autumn, but these variations are different from those in Hongkong (Xu et
al., 2015) where the highest and lowest values of HONO appeared in autumn and
spring, respectively. The important point is that the seasonality of HONO coincides
with that of $NO_x$ (or $NO_2$), which is believed to be the main precursor of HONO, in
current studies.



The HONO to $NO_x$ ratio or the HONO to $NO_2$ ratio has been used extensively in
previous research to characterize the HONO levels and to indicate the extent of
heterogeneous conversion of $NO_2$ to HONO, since it is less influenced by convection
or transport processes than the individual concentration (Lammel and Cape,
1996;Stutz et al., 2002). When a large proportion of HONO comes from direct
emissions, the value of $HONO/NO_2$ usually becomes larger, falsely implying the
strong formation of HONO from $NO_2$, however, the freshly emitted air masses
generally have the lowest $HONO/NO_x$ ratio, meaning that $HONO/NO_x$ behaves better
than $HONO/NO_2$ in a way. As shown in Fig. 1(b), the low value of $HONO/NO_x$ in
winter is attributed to heavy emissions because we see high mixing ratios of NO
during this cold season (Fig. 1c), the reasons for two peaks of $HONO/NO_x$ in spring
and summer will be discussed in sections 3.3, 3.4 and 4.

All daily changes of HONO concentration in different seasons closely resemble a
cycle in which HONO peaks in the early morning, and then decreases to the minimum
at dusk, following the diurnal trend of $NO_x$ (Fig. 2). The daily variations of HONO in
Nanjing are like those seen in other urban areas (Villena et al., 2011a;Wang et al.,
2013;Michoud et al., 2014;Lee et al., 2016), but differ from observations on the
roadside (Rappenglück et al., 2013;Xu et al., 2015). At night, the mixing ratio of
HONO increases rapidly in the first few hours and then stabilizes (in spring and
summer) or gradually climbs to its peak in the morning rush hour (in winter and
autumn). The accumulation during nighttime hours suggests a significant production
of HONO exceeding the dry deposition of HONO. As the sun rises, the HONO sink
will be strengthened by photolysis and the vertical mixing of HONO. It's clear that the
peak times varing seasonally result from different sunrise times. During the daytime,
the rate of HONO abatement is rapid before noon and then becomes progressively
until HONO concentration falling to the minimum. Given that the photolytic lifetime
of HONO is about 10-20 min in the midday (Stutz et al., 2000), the considerable
HONO concentration during daytime indicates the existence of large sources of
HONO production.






From the daily variations of the HONO to $NO_x$ ratio, we can further understand the
behavior of HONO in the atmosphere. $HONO/NO_x$ is regularly enhanced quickly
before midnight then reaches a maximum during the latter half of the night.
According to Stutz et al. (2002), the highest $HONO/NO_x$ (or $HONO/NO_2$) is defined
by the balance between production and loss of HONO at each night, the conditions
affecting the maximum ratio at nighttime will be discussed in section 3.3. What's
interesting here is the peak of the $HONO/NO_x$ ratio in the midday sun in spring,
summer and autumn, and even in winter, the ratio doesn't decline but remains
stationary before and at noon. If the HONO sources during daytime are consistent
with those at night, the minimum $HONO/NO_x$ ratios should occur at noon due to the
intense photochemical loss of HONO. Therefore, there must be additional sources of
HONO during daytime. The increase of HONO/NOx with solar radiation (e.g., UVB)
is found in both diurnal and seasonal variations, indicating that these daytime sources
have a relationship with the intensity of solar radiation. We will further discuss the
potential daytime sources of HONO in section 3.4.

The elevated mixing ratio of HONO presents an efficient reservoir of OH radicals
during daytime in Nanjing. We calculate the net OH production rate from HONO
$P_{OH}$(HONO) using Eq. (3) (Li et al., 2018). For comparison, the OH production rate
from ozone photolysis, $P_{OH}(O_3)$, is also derived from Eq. (4). Based on Alicke et al.
(2002) and Alicke (2003), only part of the $O(^1D)$ atoms, formed by the photolysis of
$O_3$ at wavelengths below 320 nm (R7), can produce OH radicals by reacting with
water (R8) in the atmosphere, so we use the absolute water concentration, which can
be derived from relative humidity and temperature, to calculate the branching ratio of
$O(^1D)$ ($\Phi_{OH}$) between R8 and R9. The reaction rate of $O(^1D)$ with $O_2$ is $4.0 \times 10^{-11}$ $cm^3$
$molecules^{-1}$ $s^{-1}$ and the reaction rate of $O(^1D)$ with $N_2$ is $3.1 \times 10^{-11}$ $cm^3$ $molecules^{-1}$
$s^{-1}$ (Seinfeld and Pandis, 2016).





$$P_{OH}(HONO) = J(HONO)[HONO] - k_{NO+OH}[NO][OH]$$
$$- k_{HONO+OH}[HONO][OH] \tag{3}$$

$$P_{OH}(O_3) = 2J(O^1D)[O_3]\phi_{OH}$$
$$\phi_{OH} = k_8[H_2O] / (k_8[H_2O] + k_9[M]) \tag{4}$$

$$O_3 + h\upsilon \rightarrow O(^1D) + O_2\,(\lambda < 320nm) \tag{R7}$$

$$O(^1D) + H_2O \rightarrow 2OH \tag{R8}$$

$$O(^1D) + M \rightarrow O(^3P) + M \quad (M\ is\ N_2\ or\ O_2) \tag{R9}$$

Fig. 3 shows that the diurnal peak of OH production rate from HONO is usually found in the late morning, caused by the combined effects of HONO concentration and its photolysis frequency, $J_{HONO}$, and the seasonal peak of $P_{OH}(HONO)$ occurs in spring for the same reason. $P_{OH}(O_3)$, coinciding with the trend of $J(O^1D)$, is highest around noon and in summer at daily and seasonal time scale respectively. Significantly, the photolysis of HONO produced more OH than that of ozone throughout the daytime in winter, spring and autumn. In summer, the contribution of HONO to OH is greater in the early morning, and, although the photolysis of ozone contributes more OH at noon, the role of HONO is considerable. Overall, the average $P_{OH}(HONO)$ during 8:00-16:00 LT is $0.90 \pm 0.27$ ppb/h, more than twice the value of $P_{OH}(O_3)$, the mean value of which is $0.41 \pm 0.25$ ppb/h. The impressive role of HONO in the atmospheric oxidizing capacity should benefit photochemical ozone production (Ding et al., 2013;Xu et al., 2017;Xu et al., 2018), new particle formation (Qi et al., 2015) and secondary aerosol formation (Xie et al., 2015;Sun et al., 2018) in Nanjing, the western YRD region.

### 3.2. Direct emissions of HONO from Combustion

As mentioned above and shown in Fig. 4(a), the similar patterns of HONO and $NO_x$, particularly sharply increasing together in the fresh plumes, in which the $NO/NO_x$ ratios are usually very high, indicate the presence of direct combustion emission of HONO, which need to be deducted when analyzing the secondary formation of



HONO. The SORPES station are influenced by air masses from both industries and
vehicles (Ding et al., 2016), the traffic emission factor investigated in other
experiments cannot be used straightly; thus, we derive the emitted HONO/NO$_x$ ratio
according the method of Xu et al. (2015), and the following five criteria are adopted
to choose fresh plumes : (a) NO$_x$>40ppbv; (b) $\triangle$NO/$\triangle$NO$_x$>0.85; (c) good
correlation between HONO and NO$_x$ (r>0.9); (d) short duration of plumes (<=2 h);
and (e) UVB<=0.01 W/m$^2$. Then, the slopes of HONO to NO$_x$ in selected plumes
were considered as the emission ratios in our study.

Within the one-year dataset, we selected 55 freshly emitted plumes satisfying the
criteria above (Table 3), of which 20 air masses were found in the morning and
evening rush hours; the derived $\triangle$HONO/$\triangle$NO$_x$ ratios vary from 0.26% to 1.91%
with a mean value of 0.79% ± 0.36%. Many factors, such as the amount of excess
oxygen; the types of fuel used (gasoline, diesel, coal); if engines are catalyst-equipped,
and if engines are well-maintained, could result in variances in these ratios.
Additionally, the rapid heterogeneous reduction of NO$_2$ on synchronously emitted BC
can also raise the value of $\triangle$HONO/$\triangle$NO$_x$ (Xu et al., 2015). For our study,
anaverage emission factor of 0.79% is deployed to evaluate the emission contribution
of HONO (Eq. 5), which is abbreviated as HONO$_{emis}$.

$$HONO_{emis} = NO_x \times 0.0079 \qquad (5)$$

$$HONO_{corr} = HONO - HONO_{emis} \qquad (6)$$


Combustion emissions contribute an average of 23% of total measured HONO
concentrations at night (Fig. 4b), with a maximum HONO$_{emis}$/HONO value of 32% in
winter and a minimum HONO$_{emis}$/HONO value of 18% in summer. We then get the
corrected observed HONO (HONO$_{corr}$) by Eq. (6) for further analysis. The slope of
the fitted line for HONO and NO$_x$ is 1.62%, higher than emission ratio 0.79% (Fig.
4a), and almost 80% of HONO is from HONO$_{corr}$ that is not affected by emissions
(Fig. 4b). These imply significant secondary formation of HONO in the atmosphere.




### 3.3. Heterogeneous conversion of NO₂ to HONO during nighttime

#### 3.3.1. The NO₂-to-HONO conversion rate (C_HONO)

In addition to emissions, heterogeneous reaction of $NO_2$ on surfaces (R4, R5) is believed to be the major formation pathway of nocturnal HONO. Thus, the $NO_2$-to-HONO conversion frequency is calculated from Eq. (5) (Alicke et al., 2002;Alicke, 2003;Wentzell et al., 2010), where $NO_2$ is adopted to scale HONO to reduce the dilution influence according to Su et al. (2008a). Similar to $HONO/NO_x$ (Fig. 2), the nighttime $HONO_{corr}/NO_2$ ratio rises from the lowest value and then reaches a quasi-stable state, meaning that $C_{HONO}$ can actually be used to assess how quickly $HONO_{corr}/NO_2$ increases to its equilibrium.

$$C_{HONO} = \frac{\dfrac{[HONO_{corr}]_{(t_2)}}{[NO_2]_{(t_2)}} - \dfrac{[HONO_{corr}]_{(t_1)}}{[NO_2]_{(t_1)}}}{t_2 - t_1} \qquad (7)$$

Following the method of Xu et al. (2015) and Li et al. (2018), 137 cases in which $HONO_{corr}/NO_2$ increased almost linearly from 18:00 to 24:00 each night are selected, and the slope fitted by the least linear regression for $HONO_{corr}/NO_2$ against time is just the conversion frequency of $NO_2$ to HONO. The derived $C_{HONO}$ vary from $0.0043 \pm 0.0017$ $h^{-1}$ in winter to $0.0066 \pm 0.0040$ $h^{-1}$ in summer, with an average value of $0.0055 \pm 0.0032$ $h^{-1}$, which is in the range (0.044-0.014 $h^{-1}$) shown by other studies in urban and suburban sites (Fig. 5). Noting that $C_{HONO}$ assumes the increase of $HONO_{corr}/NO_2$ is caused by the conversion of $NO_2$, excluding other possible sources of HONO (e.g. soil nitrite); and the computed $C_{HONO}$ is the net $NO_2$-to-HONO conversion rate since the measured $HONO_{corr}$ has already taken in to account the sinks of HONO (mainly deposition). Considering the uncertainties of $C_{HONO}$, utilizing $C_{HONO}$ directly to analyze the mechanism of HONO formation may not be appropriate, but it could be attemptable to facilitate the parameterizations for HONO production in air quality models by $C_{HONO}$.




### 3.3.2. RH dependence of HONO chemistry


It appears that $NO_2$ hydrolysis on humid surfaces (R4), having a first order
dependence on $NO_2$ (Jenkin et al., 1988;Ackermann, 2000;Finlayson-Pitts et al.,
2003), is influenced by the surface absorbed water rather than by atmospheric water
vapor (Kleffmann et al., 1998;Finlayson-Pitts et al., 2003), although the exact
mechanisms are still unknown. In the studies of Stutz et al. (2002) and Stutz et al.
(2004), the pseudo steady state of $HONO/NO_2$, where this ratio is at a maximum, is
presumed to be a balance between the production of HONO from $NO_2$ and the loss of
HONO on surfaces, and the highest $HONO/NO_2$ is determined by the ratio of the
reactive uptake coefficients for each process. Scatter plot of $HONO_{corr}/NO_2$ against
relative humidity in our study are illustrated in Fig. 5; to eliminate the influence of
other factors as for as possible, the average of the 6 highest $HONO_{corr}/NO_2$ values in
each 5% RH interval is calculated, according to Stutz et al. (2004).

The phenomenon that $HONO_{corr}/NO_2$ first increases and then decreases with an
increasing RH in Fig. 5(a) was also observed by other studies (Hao et al., 2006;Yu et
al., 2009;Li et al., 2012;Wang et al., 2013). In addition, the trend that $HONO_{corr}/NO_2$
increases roughly with RH except when RH values are greater than 95%, as shown in
Fig. 5(b), is also found in Stutz et al. (2004) and Qin et al. (2009). The dependencies
of $HONO_{corr}/NO_2$ on RH and the possible reasons or mechanisms are discussed as
follows. Even at the lowest measured RH of 18%, the absolute moisture content in the
atmosphere is still greater than $10^3$ ppm in our study, but the $HONO_{corr}/NO_2$ ratio is
quite small and remains unchanged when RH is below 45%, indicating that the $NO_2$
to HONO conversion efficiency should be determined by water covering the surfaces,
and HONO is seemingly produced on "dry" surfaces where the amount of
chemisorbed water becomes approximately independent on the water vapor levels
in dry conditions, according to Lammel (1999).

It has been reported that surfaced absorbed water depends on RH values, and the



dependences vary for different material surfaces of the ground, but generally follow
the shape of a BET isotherm (Lammel, 1999;Saliba et al., 2001;Sumner et al., 2004).
The number of mono-layers of water increases slowly from zero to 2-4, accompanied
by RH from 0 to a turning point, and the water coverage grows dramatically (up to
10-100 mono-layers) once RH exceeds the turning point (Finlayson-Pitts et al., 2003).
Fig. 5(a) shows the case where the surface for $NO_2$ converting to HONO is dominated
by the ground, the $HONO_{corr}/NO_2$ increases along with RH when RH is less than 75%,
which can be explained by the reaction of $NO_2$ to generate HONO on wet surfaces.
However, a negative correlation between $HONO_{corr}/NO_2$ and RH is found when RH is
over 75%, presumably because the rapidly growing aqueous layers of the ground
surface lead to efficient uptake of HONO and make the surface less accessible or less
reactive for $NO_2$. Hence, the RH turning point for absorbed water on ground surfaces
is perhaps around 75% for our observation, within the range of results from
experiments on various surfaces (70-80% RH) (Lammel, 1999;Saliba et al.,
2001;Sumner et al., 2004). Once RH exceeds 95%, the reaction surface is classified as
an "aqueous" surface in Lammel (1999), asymptotically approaching the state of
water droplet. Under these circumstances, the efficiency of $NO_2$ forming HONO will
be reduced since the conversion has changed from a "heterogeneous reaction" to a
"liquid reaction" (Lee and Schwartz, 1981;Cheung et al., 2000;Kleffmann et al.,
1998;Finlayson-Pitts and Pitts Jr, 1999), and the aqueous surface is found to be an
impactful sink of HONO in experimental work (Park and Lee, 1988;Becker et al.,
1996;Hirokawa et al., 2008) and in field observations(Acker et al., 2005;He et al.,
2006;Zhou et al., 2007). For the reasons mentioned above, we can see a dramatic
decline of $HONO_{corr}/NO_2$ in Fig. 5(a) and Fig. 5(b) when RH approaches 100%.

Especially deserving of mention, the constant $HONO_{corr}/NO_2$ value with RH ranging
from 75% to 95% under the condition of high $PM_{2.5}$ mass loading (Fig. 5(b)) ,
compared to the downward trend of $HONO_{corr}/NO_2$ within the same humidity range in
low $PM_{2.5}$ mass concentration (Fig. 5(a)), implies a contribution of aerosol surfaces to
the $NO_2$-HONO conversion. Since both $HONO_{corr}/NO_2$ in Fig. 5(a) and Fig. 5(b) are



affected by the ground surfaces, we can use the difference of $HONO_{corr}/NO_2$ between
Fig. 5(a) and Fig. 5(b) to represent the influence of aerosol. As the area of shadow
shown in Fig. 5(b), the aerosol-affected $HONO_{corr}/NO_2$ is positively related to RH
positively before RH reaches 95%, which is consistent with the results from
laboratory studies that the uptake coefficient of $NO_2$ to HONO ($\Upsilon_{NO2\rightarrow HONO}$) increases
with RH (Kleffmann et al., 1999;Liu et al., 2015). We will discuss the effect of
aerosol on HONO production in the next part.

*3.3.3. Impact of aerosols on HONO formation*

To further understand the heterogeneous formation of HONO on aerosol, we provide
a correlation analysis of the related $HONO_{corr}$ parameters ($HONO_{corr}$ and
$HONO_{corr}/NO_2$) with $PM_{2.5}$ when $HONO_{corr}/NO_2$ reaches the pseudo steady state each
night (3:00-6:00 LT). The convergence or diffusion processes of gases and particles
caused by the decrease or increase of the boundary layer height can also lead to a
consistent trend of $HONO_{corr}$ and $PM_{2.5}$ (Fig. 6a), while the ratio of HONO and $NO_2$
can not only remove this physical effect to a certain extent but also represent the
conversion degree of $NO_2$ to HONO, so a moderate positive correlation between
$HONO_{corr}/NO_2$ and $PM_{2.5}$ (r=0.35, p=0.01) throughout the observation period could
be more convincible (Fig. 6b). As shown by larger triangles with gray borders in Fig.
6(b), $HONO_{corr}/NO_2$ is better correlated with $PM_{2.5}$ in the months during which the
mass concentrations of $PM_{2.5}$ are higher during this 1-year measurement, generally
occurring from November to May (Fig. 1d). This finding can be explained with a law
that greater contributions of $NO_2$ heterogeneously reacting on aerosol to generate
HONO lead to better correlations between $HONO_{corr}/NO_2$ and $PM_{2.5}$. Interestingly,
this relationship can also be divided approximately into two groups by $NH_3/CO$; the
correlation is good when the value of $NH_3/CO$ is lower than 2‰, but when $NH_3/CO$ is
higher than 2‰, a poor correlation is found. We will discuss this phenomenon further
in section 4. The evidence of HONO formation on aerosol were also found in other
observations (Reisinger, 2000;Wang, 2003;Li et al., 2012;Nie et al., 2015;Hou et al.,
2016;Cui et al., 2018).




As is known, producing HONO is not the dominant sink of $NO_2$ at night, but it seems
that more $NO_2$ can be converted to HONO under conditions of heavy pollution (Fig.
7b). We discuss whether heterogeneous reactions of $NO_2$ on aerosols are able to
provide comparable HONO with our measurement by Eq. (8), where we only consider
HONO formation on particle surfaces and assume that HONO principally settles on
the ground surface, neglecting HONO loss on aerosol. $c_{NO_2}$ is the mean molecular
velocity of $NO_2$ (370 ms$^{-1}$); $[\frac{S}{V}]_{aer}$ is the surface area to volume ratio (m$^{-1}$) of aerosol;
$v_{HONO}$ is the deposition velocity of HONO, which is considered to be close to the
deposition velocity of $NO_2$ at night (Stutz et al., 2002;Su et al., 2008a); and a
approximate value of 0.1 cms$^{-1}$ is used based on the measurements from Coe and
Gallagher (1992) and Stutz et al. (2002); H is the boundary layer mixing depth, and a
value of 100 m is assumed for nighttime (Su et al., 2008a).

$$C_{HONO} = \frac{1}{4} \gamma_{NO_2 \to HONO} c_{NO_2} [\frac{S}{V}]_{aer} - \frac{v_{HONO}}{H} \frac{[HONO]}{[NO_2]} \qquad (8)$$


Considering at nighttime period with severe haze, the aerosol surface density
calculated from the particle number size distributions between 6 nm and 800 nm is
about $1.2 \times 10^{-3}$ m$^{-1}$, matched by 200 µg/m$^3$ of PM$_{2.5}$ from our observations, and the
averaged mixing ratios of HONO and $NO_2$ are 1.15 ppb and 28.4 ppb, respectively, at
night in winter (Table 2). For 30% of the measured mean winter $C_{HONO}$ (0.0013 h$^{-1}$),
the uptake coefficient of $NO_2$-to-HONO ($\gamma_{NO_2 \to HONO}$) is $6.9 \times 10^{-6}$, derived from Eq. (8),
and for all of the measured mean winter $C_{HONO}$ (0.0043 h$^{-1}$) value, the $\gamma_{NO_2 \to HONO}$ is 1.44
$\times 10^{-5}$, fitting the results from many laboratory studies which demonstrate that the
uptake coefficients of $NO_2$ ($\gamma_{NO_2}$) on multiple aerosol surfaces or wet surfaces are
mainly distributed around $10^{-5}$ with the HONO yield varying from 0.1 to 0.9 (Grassian,
2002;Aubin and Abbatt, 2007;Khalizov et al., 2010;Han et al., 2017). It is necessary



to elaborate that: (1) the ambient particles were dried with silica gel before measuring
their number size distributions, and the mass concentrations of PM$_{2.5}$ were also
measured under a system where the temperature was maintained at 30 °C; (2) the
aerosol surface was calculated using an assumption that all particles are spherically
shaped, but the particles could in fact have irregular bodies and porous structure; (3)
the particle size of both PM$_{2.5}$ and derived $[\frac{S}{V}]_{aer}$ is just a part of the total suspended
particulate matter. As described, the aerosol loading in the atmosphere is actually
underestimated in our study, thus the $\gamma_{NO_2 \rightarrow HONO}$ we derived could be the upper limit of
the uptake coefficient for NO$_2$ conversion to HONO on aerosol. In addition to
particles surfaces, other aerosol parameters such as surface water content, chemical
composition, pH value, and phase state of surfaces may also influence the
heterogeneous formation of HONO.

*3.4. Missing daytime HONO source*

498        After discussing the nocturnal formation mechanism of HONO, we now focus on

the chemistry of daytime HONO whose lifetime is only 10-20 min but whose
concentrations are still about 0.25-0.6 ppb at noon (Fig. 2). We are not certain if the
observed HONO can be provided by known mechanisms (gas phase reaction (R4) and
emissions) to date, so a budget equation of daytime HONO (Eq. 9) is utilized to
analyze its source and sinks (Su et al., 2008b;Sörgel et al., 2011). Here, dHONO/dt is
the change rate of the observed HONO. The sources rates of HONO contain the
homogeneous formation rate (P$_{NO+OH}$, R4); the combustion emission rate (P$_{emis}$); and
the unknown HONO daytime source (P$_{unknown}$). The sink rates of HONO consist of the
photolysis rate (L$_{phot}$, R1); the reaction rate of HONO with OH (L$_{HONO+OH}$, R2); and
the dry deposition rate (L$_{dep}$). T$_V$ and T$_h$ represent the vertical (T$_V$) and horizontal (T$_h$)
transport processes of HONO, which are thought to be negligible for intense radiation
and relatively homogeneous atmospheres with generally calm winds (Dillon, 2002;Su
et al., 2008b;Sörgel et al., 2011).



$$\frac{dHONO}{dt} = (P_{NO+OH} + P_{emis} + P_{unknown}) - (L_{phot} + L_{HONO+OH} + L_{dep}) + T_v + T_h \quad (9)$$


Therefore, the undiscovered daytime source of HONO ($P_{unknown}$) can be derived by Eq.
(10), which is a deformation of Eq. (9) without minor terms ($T_v$ and $T_h$) and where
dHONO/dt is substituted by $\Delta$HONO/$\Delta$t that is counted as difference between
observed HONO at two time points. The reaction rate constants of reaction 2
($k_{HONO+OH}$) and reaction 4 ($k_{NO+OH}$) are $6.0 \times 10^{-12}$ cm$^3$ molecules$^{-1}$ s$^{-1}$ and $9.8 \times 10^{-12}$
cm$^3$ molecules$^{-1}$ s$^{-1}$, respectively (Atkinson et al., 2004). The emission ratio of
HONO and NO$_x$ (HONO/NO$_x$=0.79%) obtained in section 3.2, is used to estimate
$P_{emis}$. For $L_{dep}$, the dry deposition velocity of diurnal HONO ($v_{HONO}$) is measured as 2
cms$^{-1}$ in the work of Harrison et al. (1996), and a practical mixing height of 200 m is
adopted, considering that most of the HONO cannot rise above this altitude due to
rapid photolysis (Alicke et al., 2002).

$$P_{unknown} = J(HONO)[HONO] + k_{HONO+OH}[HONO][OH] + \frac{v_{HONO}}{H}[HONO]$$
$$+ \frac{\Delta HONO}{\Delta t} - k_{NO+OH}[NO][OH] - \frac{0.79\% \times \Delta NO_x}{\Delta t} \quad (10)$$


Fig. 8 shows the average daytime HONO budget from 8:00 LT to 16:00 LT during
different seasons. The major loss route of HONO is photodecomposition ($L_{phot}$) with
an average value of 1.32 ppb/h at noontime (10:00-14:00 LT) during this observation
period, next to dry deposition ($L_{dep}$) whose mean value at the same time is 0.17 ppb/h,
and by $L_{HONO+OH}$ which is less than 3% of that of $L_{phot}$. For the sources of HONO
around noon, the average homogeneous reaction rate between NO and OH ($P_{NO+OH}$) is
0.32 ppb/h and $P_{emis}$ just gives a tiny part of HONO at a rate of 0.02 ppb/h, meaning
that most HONO comes from an unknown source whose average rate ($P_{unknown}$) is
1.13 ppb/h, contributing 76% of the production of HONO. Comparing summer data,
the mean unknown daytime source strength of HONO in Nanjing is almost at the
upper-middle level of those reported in the existing literature: 0.22 ppb/h at a rural
site of New York state, USA (Zhou et al., 2002); 0.43 ppb/h at a mountain site in





Hohenpeissenberg, Germany (Acker et al., 2006b); 0.5 ppb/h in a forest near Jülich,

Germany (Kleffmann, 2005); 0.7 ppb/h in a outskirt of Paris, France; 0.77 ppb/h in a

polluted rural area of the Pearl River Delta, China (Li et al., 2012); 0.98 ppb/h at an

urban site in Xi'an, China (Huang et al., 2017); 1.0 ppb/h in a suburban area of

Beijing, China (Yang et al., 2014); 1.7 ppb/h in an urban area of Santiago, Chile

(Elshorbany et al., 2009); 2.95 ppb/h in the urban atmosphere of Jinan, China (Li et al.,

2018); and 3.05 ppb/h at an urban site in Beijing, China (Wang et al., 2017).

The highest noontime $P_{unknown}$ value is 1.73 ppb h$^{-1}$ in spring, followed by 1.15 ppb h

$^{-1}$ in winter, 1.0 ppb h$^{-1}$ in summer and 0.77 ppb h$^{-1}$ in autumn, unliking the

seasonal variation of $NO_2$; and $P_{unknown}$ shows an increase towards noon, this

production rate is higher before noon than after noon, which is also distinguished

from the diurnal pattern of $NO_2$. These results indicate that the production of daytime

HONO is different from the heterogeneous formation from $NO_2$ at night. Hence, we

perform a correlation analysis between noontime $P_{unknown}$ and related parameters to

determine the potential unknown daytime source of HONO (Table 4). $P_{unknown}$ is

better correlated with $NO_2$*UVB than with $NO_2$ or UVB alone in winter, spring and

autumn (p=0.05), perhaps associated with the photo-enhanced converting from $NO_2$

to HONO (George et al., 2005;Stemmler et al., 2006;Stemmler et al., 2007), and this

is the reason for $P_{unknown}$ normalized by $NO_2$ following the steps of UVB, showing a

peak around noontime in different seasons (Fig. 8). The average value of $P_{unknown}$

normalized by $NO_2$ is 0.1 h$^{-1}$, over 18 times greater than the nighttime conversion rate

(0.0055 h$^{-1}$), also implying that $P_{unknown}$ cannot be explained by the $NO_2$-to-HONO

mechanism at night. Assuming that the height of a well-mixed boundary layer around

remain constant for each day, UVB*$NO_2$ and UVB*$NO_2$*$PM_{2.5}$ could be proxies for

photo-induced heterogeneous reactions of $NO_2$ on ground and aerosol surfaces,

respectively. In winter and spring, the correlation numbers of $P_{unknown}$ with UVB*$NO_2$

are similar to those of $P_{unknown}$ with UVB*$NO_2$*$PM_{2.5}$, while if we only consider

UVB*$NO_2$ instead of UVB*$NO_2$*$PM_{2.5}$ in summer and autumn, the correlations





significantly increase (p=0.05). We cannot be sure which surfaces (ground or aerosol)
are more important to the hypothetical photo-heterogeneous reaction of $NO_2$ based on
the present study. The photolysis of particulate nitrates ($NO_3^-$) as a source of HONO
(Ye et al., 2016;Ye et al., 2017) cannot be determined if it is momentous in our study,
since the correlation between $P_{unknown}$ and $UVB*NO_2$ isn't superior to the correlation
between $P_{unknown}$ and UVB multiplied by $PM_{2.5}$ or other aerosol compounds. The
comparisons of correlation coefficients shown above follow the method provided by
Meng et al. (1992). Overall, it seems that the sealed source of daytime HONO is
optically controlled, although we are not sure what the actual mechanism is.

*4. Estimation of the contribution from different sources*

From this and previous studies, we can conclued that not only the concentration of
ambient HONO but also the sources of HONO have temporal and spatial patterns,
which is supposed to be considered in model studies. Nocturnal HONO is selected to
discuss the monthly variations of HONO sources in detail without the uncertainties of
daytime HONO formation and the influences of HONO photolysis. The
heterogeneous reaction of $NO_2$ on aerosol produces a considerable portion of HONO
in relatively polluted months (Dec.-May), but contributes very little less than nothing
in clean months (Jun.-Oct.), as seen in section 3.3.3. Coincidentally, direct emissions
from burning processes of HONO decrease from their peak values from winter to
summer (section 3.2). However, the monthly averaged ratios of HONO and $NO_x$ are
highest in summer , which conflicts with two sources mentioned above.

As is known, higher $NO_2$-to-HONO conversion level or other $NO_x$-independent
sources can cause an increase in the HONO/$NO_x$ ratio. For the case of a mostly
constant surface with low reactivity due to the long-term exposure to oxidizing gases
and radiation, the yield of nighttime HONO from $NO_2$ reacting on ground surfaces
could be imprecisely assumed to be unchanged. Thus, soil nitrite formed through
microbial activities, especially nitrification by ammonia-oxidizing bacteria
($NH_4^+ \rightarrow NO_2^-$) (Su et al., 2011;Oswald et al., 2013), is adopted to be an source for



atmospheric HONO in this study, considering the nearby presence of some grassland
and natural vegetation mosaics. Although we do not directly measure HONO
emissions from soil, the observed ammonia can represent its monthly average
intensity, based on the following hypothesis: the dominant source of $NH_3$ is from soil,
especially from fertilizers ($NH_4^+ \rightarrow NH_3$) for a good correlation between ammonia and
temperature in the site (r=0.63, p=0.01), omitting the contributions of livestock to
$NH_3$ since there is only a small poultry facility within 10 km of this site (Meng et al.,
2011;Huang et al., 2012;Behera et al., 2013). Combustion sources (vehicles, industry,
biomass burning) should contribute only a fraction of $NH_3$ seeing that $NH_3$ is not
related to NOx or CO in our study. Moreover, the release of both HONO and $NH_3$
depend on the strength of microbial activities, fertilizing amount, and soil properties
(e.g., temperature, acidity and water content of soil). Although the processes of
HONO and $NH_3$ emission from soil may not be completely synchronized, the
seasonal patterns for each should be consistent.

Until now, we can separate the sources of HONO into four parts: (1) combustion
emissions from vehicles and industries ($HONO_{emi}$) with a constant emitted
HONO/NOx ratio of 0.79%; (2) conversion of $NO_2$ to HONO on the ground surfaces
($HONO_{grd}$) with a constant but unknown yield $x_1$; (3) conversion of $NO_2$ to HONO on
aerosol surfaces ($HONO_{aer}$) with a $PM_{2.5}$-dependent yield ($HONO_{are}/NO_2$); and (4)
emission from soil ($HONO_{soi}$), expressed by corrected $NH_3$ multiplied by an unknown
coefficient $x_2$. The corrected $NH_3$ is obtained by subtracting combustion emission
from total observed ammonia. Ammonia from combustion is found to be proportional
to simultaneous CO (Meng et al., 2011;Chang et al., 2016), and a proportion of 0.3%,
which is in the lower quantile of the $NH_3$/CO ratios in fresh air masses (for hourly
data: $NO/NO_x > 0.75$; UVB=0; temperature<5℃) is used from our measurements.
Substituting monthly average values of measured HONO, $NO_2$, $PM_{2.5}$, $NH_3$, and CO
into Eq. (9) by assuming that $HONO_{tot}$ is equal to $HONO_{obs}$, we can get an
overdetermined system of equations with 11 equations with 2 unknowns (excluding
means of related parameters from February), finally achieving an approximate





solution ($x_1$=1.89%, $x_2$=1.62%) by the method of ordinary least squares.

Fig. 9 shows that an average of 36% of HONO is produced heterogeneously on
ground surfaces without perceptible temporal variations, but the contribution of this
source is overtaken by $NO_2$ converting to HONO on aerosols in January
(approximately 40% of HONO), and was exceeded by soil emission in July and
August (approximately 40% of HONO). The seasonal variations of HONO from
different pathways at night indicate that short-term observations may just capture a
small part of the total picture when exploring the source mechanisms of HONO. The
total HONO concentration ($HONO_{tot}$) is the sum of derived HONO from the four
sources listed above. The good correlation between $HONO_{tot}$ and $HONO_{obs}$ and the
low mean normalized error of $HONO_{tot}$ to $HONO_{obs}$ indicate that our assumption
regarding nocturnal HONO sources is reasonable. It should be noted that the slope of
the linearly fitted line between $HONO_{corr}/NO_2$ and $PM_{2.5}$ in spring (r=0.74,
slope=0.68‰) is much higher than that in winter (r=0.60, slope=0.20‰), but we just
use a mean slope of 0.26‰ to evaluate aerosol effects throughout the year, this may
be why our method underestimates HONO in March and April and overestimates
HONO in January, and revealing that the mass concentration of $PM_{2.5}$ is not the only
factor affecting formation of HONO on aerosols. Besides, lack consideration of the
impact of RH and temperature on $NO_2$-to-HONO conversion and of seasonal
variations in ground surface properties, uncertainties of $NO_2$-to-HONO conversion
mechanisms and of combustion HONO emissions, and lack direct observation for soil
emitted HONO, could all result in the bias between $HONO_{tot}$ and $HONO_{obs}$, so more
studies on the detailed mechanism of various HONO sources need to be performed.



$$\frac{[HONO_{grd}]}{[NO_2]} = x_1$$

$$\frac{[HONO_{aer}]}{[NO_2]} = 0.26\text{‰} \times [PM_{2.5}]$$

$$\frac{[HONO_{emi}]}{[NO_x]} = 0.79\% \tag{11}$$

$$\frac{[HONO_{soi}]}{[NH_3] - 0.3\% \times [CO]} = x_2$$

$$HONO_{tot} = HONO_{emi} + HONO_{soi} + HONO_{grd} + HONO_{aer}$$

### 5. Conclusions

Continuous field measurement of HONO over one year was conducted at SORPES station in Nanjing in the western Yangtze River Delta (YRD), China, from December, 2017 to December, 2018. The observed seasonal average of HONO concentrations are in the range of 0.45-1.04 ppb, which are comparable to those in other urban or suburban regions and appears to be of vital importance to atmospheric oxidation as the photolysis rate of HONO is over 2 times that of ozone at daytime. HONO and $NO_x$ have coincident monthly variations peaking in December and decreasing to the lowest value in August, and have similar diurnal pattern with the highest value in the early morning and a low point before dusk, both indicating that $NO_x$ is a crucial precursor of HONO.

Combustion emissions contribute an average of 23% to nocturnal HONO concentrations, with an average emission ratio $\triangle HONO/\triangle NO_x$ of 0.79%. During the nighttime, the dominant source of RH-dependent HONO could be the heterogeneous reaction of $NO_2$ on wet ground or aerosol surfaces with a mean estimated conversion rate of 0.0055 $h^{-1}$. During the daytime, a missing HONO source with an average strength of 1.13 ppb $h^{-1}$ was identified around noon, contributing more than 75% of the production of HONO and seeming to be photo-enhanced. HONO released from soil is adopted to discuss the seasonal changes of nocturnal HONO, and can contribute 40% to HONO during summer. Ground formation



provides a major part of HONO at roughly constant proportion of 36%. The uptake of
$NO_2$ on aerosol surface could generate the greatest amount of HONO during heavily
polluted periods (e.g. January). Our results draw a complete picture of the sources of
HONO during different seasons, and demonstrated the needs of long-term and
comprehensive observations to improve the understanding of HONO chemistry.
**Author contribution**
W.N. and A.D. designed the study; Y.L. and W.N. wrote the manuscript; Y.L., Z.X.
and R.X. collected the HONO data and contributed to the data analysis; T.W., Y.L.,
L.W. and X.C. collected other related data, e.g. $NH_3$, $NO_x$ and $PM_{2.5}$.
**Acknowledgments**
This work was mainly funded by the National Key R&D Program of China
(2016YFC0202000 and 2016YFC0200500), and the National Natural Science
Foundation of China (NSFC) project (D0512/41675145 and D0510/41605098). Data
analysis was also supported by other NSFC projects (D0512/41875175 and
D0510/41605098).

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



**Tables**

**Table 1.** Sources and sinks for nitrous acid (HONO) in the troposphere.

| Budget | Occurrence | Pathways | Abbr. |
|--------|-----------|----------|-------|
| Sinks | Only daytime | $HONO + h\upsilon \xrightarrow{320-400\,nm} OH + NO$ | R1 |
| | Mainly daytime | $HONO + OH \rightarrow NO_2 + H_2O$ | R2 |
| | All day | Deposition/heterogeneous loss on aerosol | / |
| Sources | Mainly daytime | $NO + OH \xrightarrow{M} HONO$ | R3 |
| | Mainly nighttime | $2NO_{2(g)} + H_2O_{(ads)} \xrightarrow{surf} HONO_{(g)} + HNO_{3(ads)}$ | R4 |
| | Mainly daytime | $NO_{2(g)} + HC_{red} \xrightarrow{surf} HONO_{(g)} + HC_{ox}$ | R5 |
| | Only daytime | $HNO_3 / NO_3^- + h\upsilon \xrightarrow{surf} HONO / NO_2^- + O$ | R6 |
| | All day | Release of soil nitrite | / |
| | All day | Combustion emission (fossil and biomass) | / |



**Table 2.** Overview of the measured HONO and NOx levels in Nanjing and comparison with other urban or suburban sites.

| Location | Date | HONO(ppb) | | NO₂(ppb) | | NOx(ppb) | | HONO/NO2 | | HONO/NOx | | Ref |
|---|---|---|---|---|---|---|---|---|---|---|---|---|
| | | Night | Day | Night | Day | Night | Day | Night | Day | Night | Day | |
| Rome(Italy) | May-Jun 2001 | 1.00 | 0.15 | 27.2 | 4.0 | 51.2 | 4.2 | 0.037 | 0.038 | 0.020 | 0.036 | 1 |
| Kathmandu(Nepal) | Jan-Feb 2003 | 1.74 | 0.35 | 17.9 | 8.6 | 20.1 | 13.0 | 0.097 | 0.041 | 0.087 | 0.027 | 2 |
| Tokto(Japan) | Jan-Feb 2004 | 0.80 | 0.05 | 31.8 | 18.2 | 37.4 | 26.3 | 0.025 | 0.003 | 0.021 | 0.002 | 3 |
| Santiago(Chile) | Mar 2005 | 3.00 | 1.50 | 30.0 | 20.0 | 200.0 | 40.0 | 0.100 | 0.075 | 0.015 | 0.038 | 4 |
| Mexico City(Mexico) | Mar 2006 | / | 0.43 | / | 28.4 | / | 44.8 | / | 0.015 | /. | 0.010 | 5 |
| Houston(USA) | Sep 2006 | 0.50 | 0.10 | 20.0 | 10.0 | / | / | 0.025 | 0.010 | / | / | 6 |
| Shanghai(China) | Oct 2009 | 1.50 | 1.00 | 41.9 | 30.0 | / | / | 0.038 | 0.032 | / | / | 7 |
| Hongkong(China) | Aug 2011 | 0.66 | 0.70 | 21.8 | 18.1 | 29.3 | 29.3 | 0.031 | 0.042 | 0.025 | 0.028 | 8 |
| | Nov 2011 | 0.95 | 0.89 | 27.2 | 29.0 | 37.2 | 40.6 | 0.034 | 0.030 | 0.028 | 0.021 | |
| | Feb 2012 | 0.88 | 0.92 | 22.2 | 25.8 | 37.8 | 48.3 | 0.036 | 0.035 | 0.025 | 0.020 | |
| | May 2012 | 0.33 | 0.40 | 14.7 | 15.0 | 19.1 | 21.1 | 0.022 | 0.030 | 0.019 | 0.022 | |
| Beijing(China) | Oct–Nov 2014 | 1.75 | 0.93 | 37.6 | 35.3 | 94.5 | 53.4 | 0.047 | 0.026 | 0.019 | 0.017 | 9 |
| Xi'an(China) | Jul–Aug 2015 | 0.51 | 1.57 | 15.4 | 24.7 | / | / | 0.033 | 0.062 | / | / | 10 |
| Jinan(China) | Sep–Nov 2015 | 0.87 | 0.66 | 25.4 | 23.2 | 38.0 | 37.5 | 0.049 | 0.034 | 0.034 | 0.022 | 11 |
| | Dec 2015-Feb 2016 | 2.15 | 1.35 | 41.1 | 34.6 | 78.5 | 64.8 | 0.056 | 0.047 | 0.034 | 0.031 | |
| | Mar–May 2016 | 1.24 | 1.04 | 35.8 | 25.8 | 47.3 | 36.0 | 0.046 | 0.052 | 0.035 | 0.041 | |
| | Jun–Aug 2016 | 1.20 | 1.01 | 22.5 | 19.0 | 29.1 | 25.8 | 0.106 | 0.079 | 0.060 | 0.049 | |
| Nanjing(China) | Nov 2017-Nov 2018 | 0.80 | 0.57 | 18.9 | 13.9 | 24.9 | 19.3 | 0.045 | 0.044 | 0.041 | 0.036 | this |
| | Dec-Feb(winter) | 1.15 | 0.92 | 28.4 | 23.1 | 45.5 | 37.7 | 0.040 | 0.038 | 0.029 | 0.025 | study |
| | Mar–May(spring) | 0.76 | 0.59 | 17.4 | 12.9 | 19.1 | 15.9 | 0.048 | 0.049 | 0.046 | 0.042 | |
| | Jun–Aug(summer) | 0.56 | 0.34 | 12.5 | 7.7 | 13.5 | 9.1 | 0.048 | 0.051 | 0.046 | 0.045 | |
| | Sep–Nov(autumn) | 0.81 | 0.51 | 18.9 | 13.4 | 25.1 | 17.7 | 0.044 | 0.035 | 0.039 | 0.029 | |

1: Acker et al. (2006a); 2: Yu et al. (2009); 3: Kanaya et al. (2007); 4: Elshorbany et al. (2009); 5: Dusanter et al. (2009); 6: Wong et al. (2011); 7: Bernard et al. (2016); 8: Xu et al. (2015); 9: Tong et al. (2015); 10: Huang et al. (2017); 11: Li et al. (2018)



**Table 3.** the emission ratios ΔHONO/ΔNO$_x$ in 55 selected fresh plumes emitted, r is the correlation coefficient between ΔHONO and ΔNO$_x$.

| Start Time | Duration(min) | △NO/△NO$_x$ | r | △HONO/△NO$_x$(%) |
|---|---|---|---|---|
| 11/15/2017 19:50 | 20 | 0.88 | 1.00 | 0.51 |
| 11/15/2017 20:50 | 30 | 1.00 | 0.97 | 0.42 |
| 11/15/2017 21:40 | 30 | 1.05 | 0.91 | 0.59 |
| 11/20/2017 21:50 | 110 | 0.91 | 0.90 | 0.79 |
| 11/21/2017 03:10 | 70 | 1.06 | 0.97 | 0.33 |
| 11/22/2017 05:40 | 30 | 1.14 | 0.93 | 0.35 |
| 11/23/2017 06:00 | 20 | 1.10 | 1.00 | 0.26 |
| 11/23/2017 19:30 | 110 | 1.09 | 0.92 | 0.67 |
| 11/24/2017 01:50 | 20 | 1.23 | 0.94 | 0.63 |
| 11/24/2017 17:20 | 20 | 0.91 | 0.98 | 0.48 |
| 11/28/2017 23:30 | 40 | 1.07 | 0.91 | 0.82 |
| 12/01/2017 01:10 | 120 | 1.07 | 0.91 | 0.64 |
| 12/03/2017 04:40 | 20 | 1.00 | 0.97 | 0.29 |
| 12/03/2017 23:00 | 30 | 1.12 | 0.93 | 1.91 |
| 12/07/2017 01:40 | 50 | 1.12 | 0.96 | 0.68 |
| 12/07/2017 05:40 | 20 | 0.85 | 0.93 | 0.42 |
| 12/07/2017 06:40 | 20 | 0.93 | 0.98 | 0.63 |
| 12/08/2017 19:40 | 20 | 1.08 | 0.99 | 0.98 |
| 12/08/2017 23:30 | 120 | 1.10 | 0.93 | 0.74 |
| 12/09/2017 20:20 | 120 | 0.89 | 0.90 | 0.74 |
| 12/11/2017 04:10 | 30 | 1.13 | 0.92 | 0.93 |
| 12/11/2017 07:10 | 20 | 1.13 | 0.94 | 0.97 |
| 12/17/2017 19:40 | 120 | 1.04 | 0.96 | 1.02 |
| 12/18/2017 03:10 | 90 | 1.09 | 0.94 | 0.59 |
| 12/20/2017 23:50 | 40 | 1.15 | 0.95 | 0.97 |
| 12/21/2017 01:20 | 40 | 1.21 | 0.90 | 0.98 |
| 12/21/2017 03:20 | 40 | 1.28 | 0.95 | 1.52 |
| 12/23/2017 07:10 | 30 | 0.96 | 0.97 | 1.09 |
| 12/23/2017 17:10 | 60 | 1.08 | 0.97 | 1.11 |
| 12/25/2017 02:00 | 110 | 1.05 | 0.91 | 0.81 |
| 12/26/2017 20:20 | 20 | 1.05 | 0.99 | 0.96 |
| 12/28/2017 22:40 | 100 | 1.02 | 0.92 | 1.29 |
| 01/10/2018 22:20 | 120 | 1.02 | 0.95 | 1.05 |
| 01/20/2018 23:50 | 120 | 1.20 | 0.93 | 1.36 |
| 01/21/2018 03:00 | 70 | 1.01 | 0.98 | 0.66 |




| 01/21/2018 19:20 | 50  | 1.03 | 0.98 | 0.38 |
|------------------|-----|------|------|------|
| 01/30/2018 23:00 | 40  | 1.09 | 0.94 | 0.67 |
| 01/31/2018 04:10 | 70  | 1.16 | 0.90 | 0.67 |
| 01/31/2018 18:30 | 20  | 0.96 | 0.96 | 0.74 |
| 03/22/2018 03:30 | 110 | 0.87 | 0.94 | 0.44 |
| 04/10/2018 21:40 | 110 | 0.91 | 0.94 | 1.02 |
| 04/15/2018 20:50 | 50  | 0.94 | 0.97 | 1.47 |
| 04/27/2018 03:10 | 20  | 1.03 | 0.99 | 0.46 |
| 05/12/2018 19:20 | 30  | 0.88 | 0.92 | 1.21 |
| 05/30/2018 02:50 | 30  | 0.88 | 1.00 | 1.50 |
| 07/04/2018 02:20 | 20  | 0.91 | 1.00 | 0.42 |
| 09/28/2018 19:30 | 20  | 0.94 | 0.96 | 1.37 |
| 09/28/2018 21:40 | 40  | 0.99 | 0.98 | 0.80 |
| 09/29/2018 23:40 | 20  | 0.92 | 1.00 | 0.33 |
| 10/11/2018 01:20 | 120 | 0.96 | 0.92 | 0.65 |
| 10/22/2018 21:50 | 20  | 0.94 | 0.99 | 0.42 |
| 10/26/2018 22:50 | 110 | 1.07 | 0.90 | 0.92 |
| 10/27/2018 21:00 | 30  | 1.00 | 0.91 | 0.63 |
| 10/28/2018 00:00 | 70  | 0.99 | 0.91 | 0.58 |
| 11/21/2018 04:10 | 70  | 0.95 | 0.94 | 0.51 |



**Table 4.** Correlations of $P_{unknown}$ against various parameters.

| Parameters | Winter | | Spring | | Summer | | Autumn | |
|---|---|---|---|---|---|---|---|---|
| | r | N | r | N | r | N | r | N |
| $NO_2$ | 0.45 | 254 | 0.35 | 310 | -0.04 | 423 | 0.23 | 381 |
| $PM_{2.5}$ | 0.41 | 254 | 0.45 | 310 | 0.18 | 423 | 0.31 | 381 |
| $NO_3^-$ | 0.42 | 245 | 0.45 | 298 | -0.03 | 409 | 0.21 | 373 |
| $SO_4^{2-}$ | 0.33 | 236 | 0.29 | 298 | 0.12 | 413 | 0.21 | 368 |
| $NH_4^+$ | 0.39 | 245 | 0.41 | 301 | 0.05 | 416 | 0.23 | 359 |
| RH | 0.02 | 254 | -0.34 | 310 | -0.42 | 423 | -0.16 | 381 |
| UVB | 0.27 | 254 | 0.43 | 310 | 0.51 | 423 | 0.42 | 381 |
| $NO_2*PM_{2.5}$ | 0.43 | 254 | 0.46 | 310 | 0.06 | 423 | 0.25 | 381 |
| $NO_2*NO_3^-$ | 0.43 | 245 | 0.45 | 298 | -0.04 | 409 | 0.19 | 373 |
| $NO_2*SO_4^{2-}$ | 0.41 | 236 | 0.40 | 298 | 0.05 | 413 | 0.21 | 368 |
| $NO_2*NH_4^+$ | 0.42 | 245 | 0.45 | 301 | 0.01 | 416 | 0.20 | 359 |
| $UVB*NO_2$ | 0.65 | 254 | 0.67 | 310 | 0.48 | 423 | 0.59 | 381 |
| $UVB*PM_{2.5}$ | 0.58 | 254 | 0.64 | 310 | 0.50 | 423 | 0.64 | 381 |
| $UVB*NO_3^-$ | 0.55 | 245 | 0.59 | 298 | 0.24 | 409 | 0.45 | 373 |
| $UVB*SO_4^{2-}$ | 0.43 | 236 | 0.51 | 298 | 0.42 | 413 | 0.34 | 368 |
| $UVB*NH_4^+$ | 0.51 | 245 | 0.57 | 301 | 0.32 | 416 | 0.49 | 359 |
| $NO_2*UVB*PM_{2.5}$ | 0.59 | 254 | 0.65 | 310 | 0.37 | 423 | 0.49 | 381 |





**Figures**


**Fig. 1.** Monthly variations of (a) HONO, (b) HONO/NO$_x$, (c) NO$_x$, (d) PM$_{2.5}$, (e) RH and (f)T.
The solid bold lines are median values, the markers indicate mean values, and the shaded areas
represent percentiles of 75% and 25%. In (a) and (b), values in February are linearly interpolated
based on the data from the months before and after, since there were only few days when HONO
was observed in February. In (c), the shaded area is colored by the 25$_{th}$ to the 75$_{th}$ percentiles of
NO.



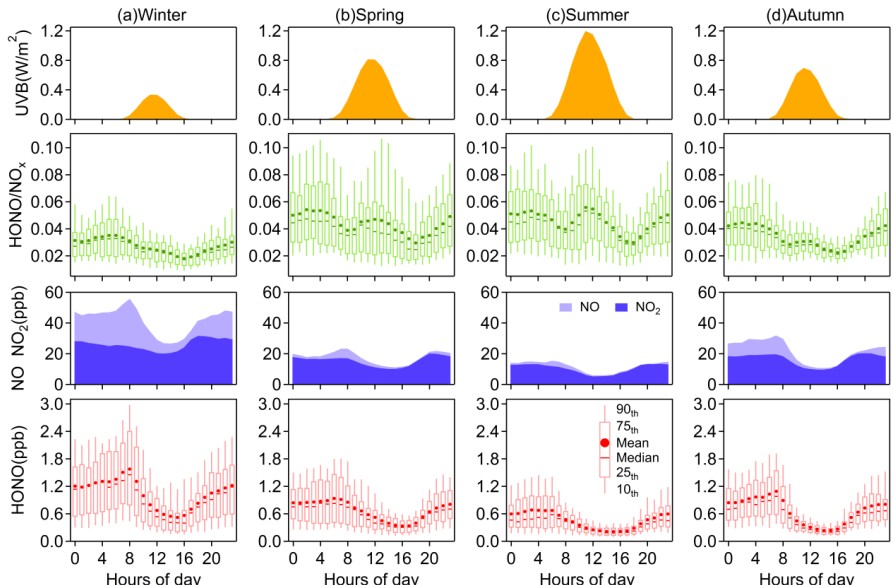

**Fig. 2.** Diurnal variations of HONO, NO, $NO_2$, HONO/$NO_x$, UVB in (a) winter, (b) spring, (c)summer, (d)autumn. NO, $NO_2$ and UVB values are displayed as their mean concentrations.



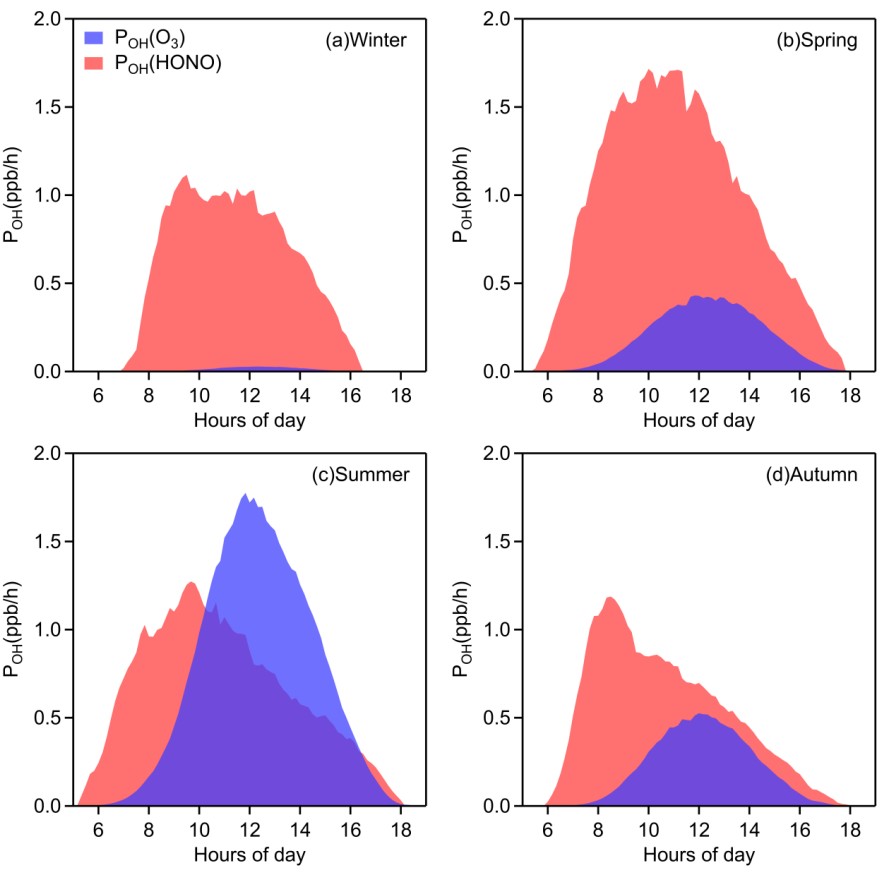


**Fig. 3.** Average OH production rates from photolysis of HONO and $O_3$ in (a)winter, (b)spring, (c)summer, and (d)autumn.











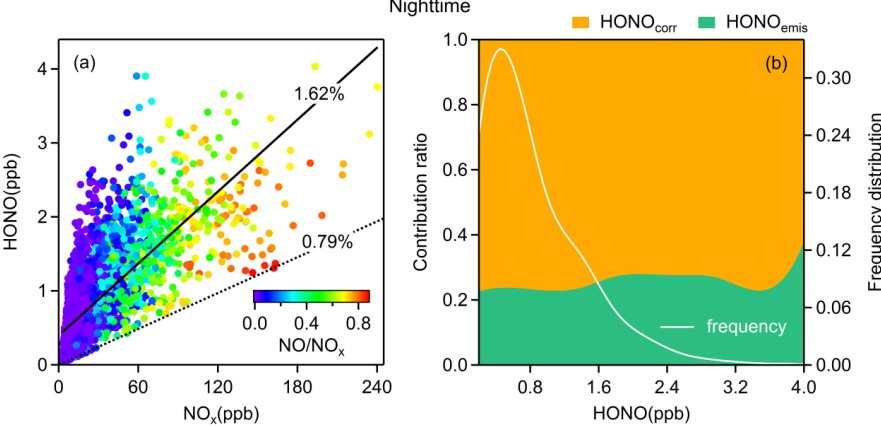


**Fig. 4.** (a) The relationship between HONO and $NO_x$ colored by $NO/NO_x$. The dotted line is the
emission ratio derived in this study and the solid line is obtained from simple linear fitting; (b)
average emission contribution ratios for different concentrations of HONO and the frequency
distribution of HONO concentrations. Both (a) and (b) are nighttime values.






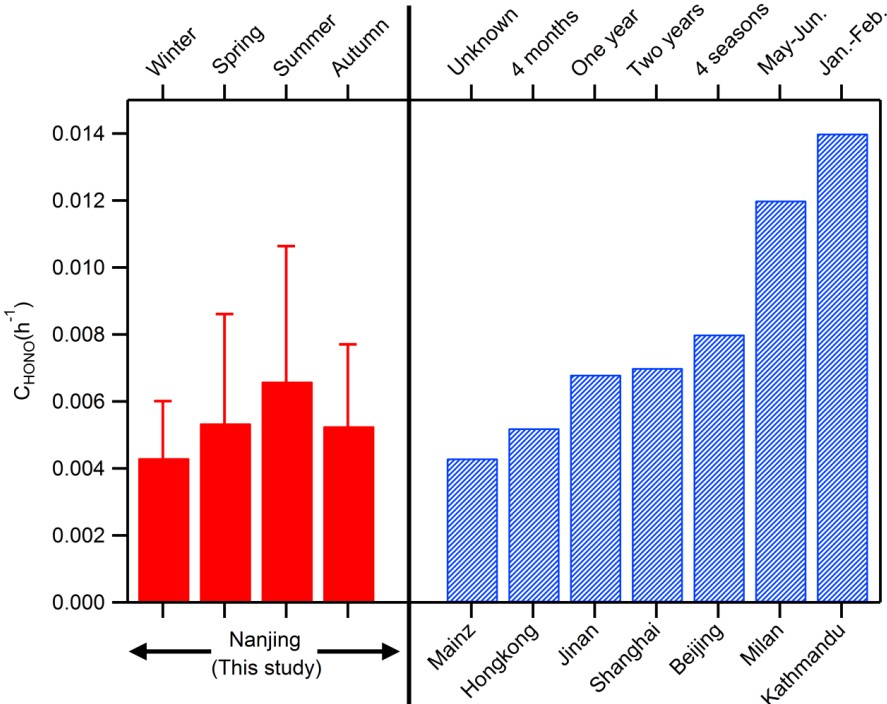


**Fig. 5.** Comparison of observed $NO_2$ to HONO conversion rates in cities: Nanjing (this study); Mainz (Lammel, 1999); Hongkong (Xu et al., 2015); Jinan (Li et al., 2018); Shanghai (Wang et al., 2013); Beijing (Wang et al., 2017); Milan (Alicke et al., 2002); and Kathmandu (Yu et al., 2009).



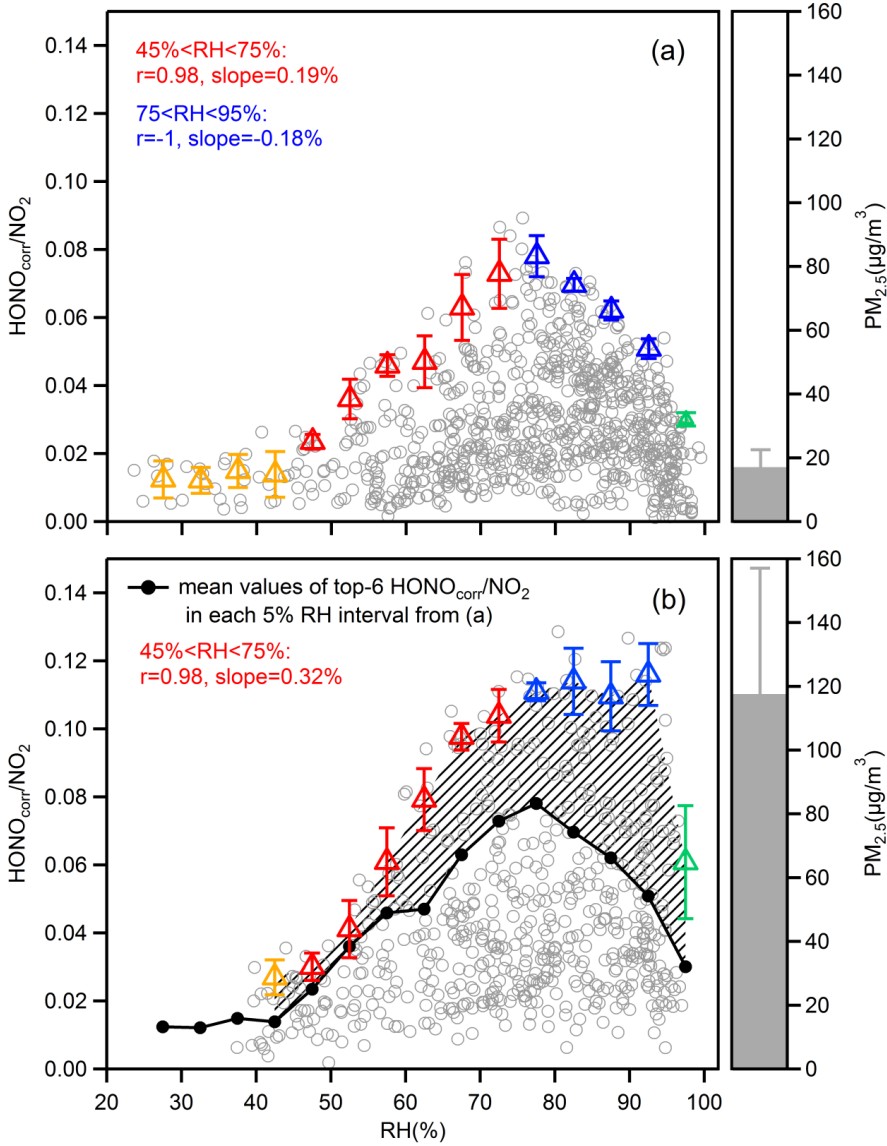

**Fig. 6.** Scatter plot of HONO_corr/NO₂ and RH during nighttime, separating the data into (a) clean hours (hourly mean PM$_{2.5}$<25μg/m³) and (b) pollution hours (hourly mean PM$_{2.5}$>75μg/m³). Triangles are the averaged top-6 HONO_corr/NO2 in each 5% RH interval, and the error bars are the standard deviations. The overall average concentrations of PM$_{2.5}$ in (a) and (b) are shown to the right of the figure.



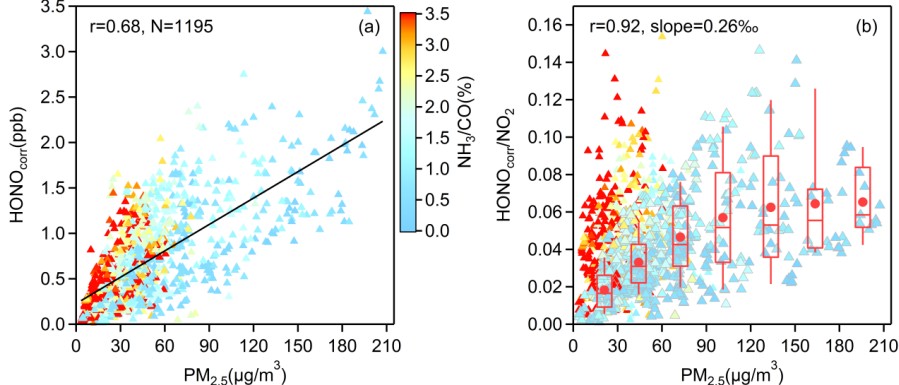

**Fig. 7.** The correlation between HONO$_{corr}$ and PM$_{2.5}$ (a), and the correlation between HONO$_{corr}$/NO$_2$ and PM$_{2.5}$ (b), all scatters come from the time (3:00-6:00 LT) when HONO$_{corr}$/NO$_2$ reaches the pseudo steady state each night and are colored by NH$_3$/CO. In (b), the larger triangles with gray borders, depict the measured data from November to May, and the boxplot in each 30 µg/m$^3$ interval of PM$_{2.5}$ is illustrated according to the same data, the red box boundaries represent interquartile range, the whiskers represent the 10%–90% percentile range, the horizontal red lines represent median values and the red markers represent mean values. The correlation coefficient and the slope of the linearly fitted line in (b) are derived from the average HONO$_{corr}$/NO$_2$ and average PM$_{2.5}$ in each box.



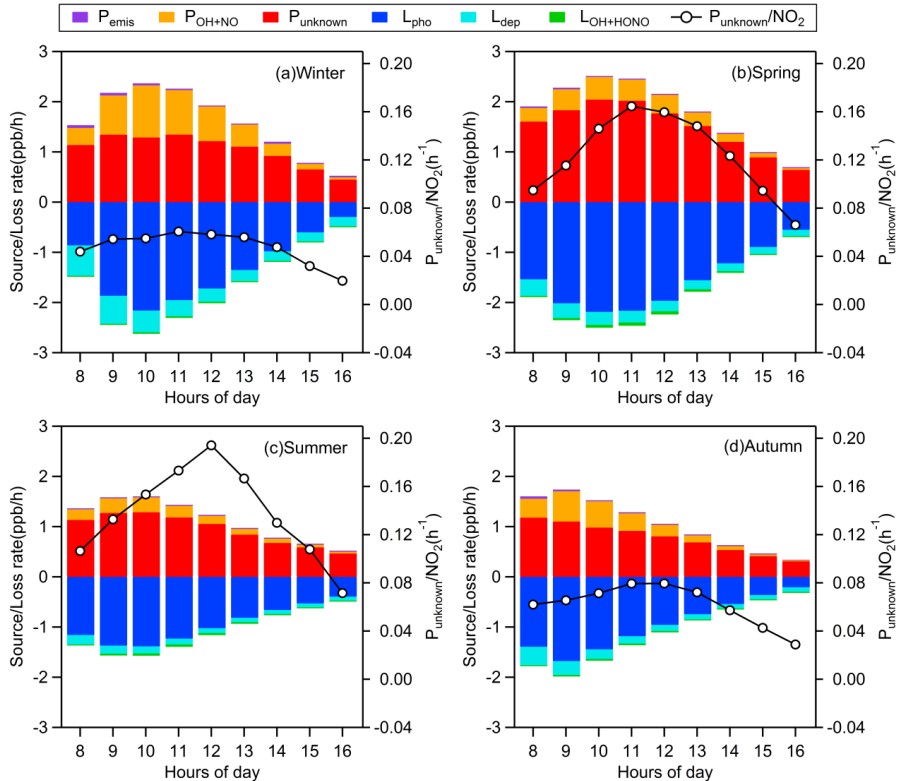

**Fig. 8.** Average daytime HONO budget and the missing source strength ($P_{unknown}$) normalized by $NO_2$ in (a) winter, (b) spring, (c) summer, and (d) autumn



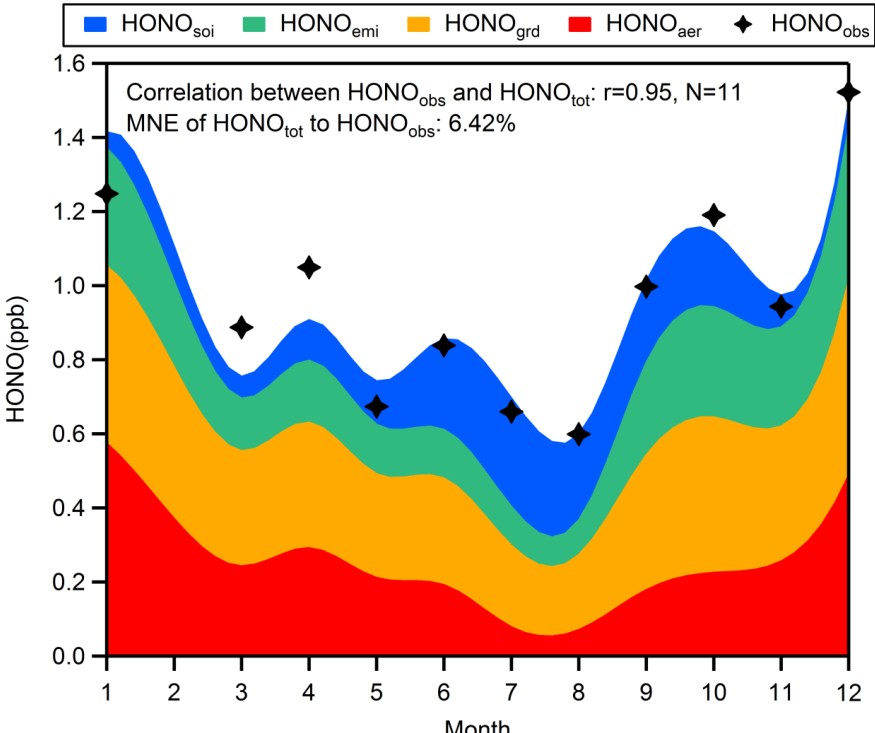

**Fig. 9.** Seasonal variations of 4 sources of mean HONO at night (3:00-6:00 LT). The mean normalized error (MNE) of $HONO_{tot}$ to $HONO_{obs}$ is 6.42%.