# Peer review of "Semi-quantitative understanding of source contribution to nitrous acid (HONO) based on 1-year continuous observation at the SORPES station in eastern China"

_Atmospheric Chemistry and Physics, 2019_

## Referee Comment (RC1) · Anonymous Referee #1 · 28 May 2019

This paper reports year round measurements of HONO in Nanjing , Eastern China. The effect of direct emissions of HONO is calculated by looking at fresh plumes and production of HONO from heterogeneous reaction of NO2 on aerosol surfaces was speculated upon using nighttime HONO and RH measurements. A HONO budget was calculated, along with a missing HONO source. The effect of the measured HONO on the OH budget was also described. Understanding the role of HONO is crucial for the understanding of oxidation chemistry, especially in the urban environment therefore this study is important work that should be published. There are very few long term measurements of HONO in the literature, with most studies being done in short term campaigns. The analysis here is a reasonable attempt at understanding the role of

[Figure]

HONO, albeit with a fairly limited set of supporting measurements. It is within scope of ACP and I recommend publication subject to completion of the following modifications.

General comments: No OH measurements were available during the measurements period so the authors have calculated OH concentrations for their analysis (P7). They use the work of Rohrer and Berresheim that correlates OH with J(O1D). I find this a strange choice of literature to use as it was based on work in a very different environment. I believe there are numerous measurements of OH taken within the PRD that would be a more relevant way to infer OH concentrations for this study.

The authors present a calculation of the effect of HONO on OH formation and compare it to formation by O3 photolysis. This study needs expanding a little. OH production from the HO2 + NO reaction would likely be the largest source in such an environment as this study. If the authors want to look at HOx radical formation then they should also make some comment about the effect of other sources such as HCHO photolysis to form HO2 and O3 + alkene reactions. I realise they may not have the supporting measurements to do this accurately but some mention should be made of it.

Some comment should be made as to how much the 'missing HONO' source contributes to OH. This is important in terms of understanding how much models might be underestimating OH by not having all the HONO sources in them.

Following on from this, some mention should be made about how various air quliaty forecasting or regional models treat HOHO and how adding in the 'unconventional' sources might affect oxidation chemistry.

Minor comments: P5 line 150: Can the aithors justify that the measurement is 'interference free'? It was my understanding that the LOPAP instrument is subject to interference from other nitrate species. Please expand this.

P8 line 244: 'in the' should be 'at'

P9 Line 264: The authors describe HONO as an efficient reservoir of OH radicals. I'm

not totally convinced this is the correct way to describe it. A reservoir suggests a long lived species that enables transport of OH radicals. I would have thought the lifetime of HONO would be very short, maybe the authors could comment on this.
* * *

---

## Referee Comment (RC2) · Anonymous Referee #3 · 10 Jun 2019

This manuscript describes continuous ambient measurements of HONO, NO, NO2, and PM2.5 at the SORPES station in Nanjing (eastern China) from November 2017 to November 2018. The main conclusions are:

1. Seasonal average HONO concentrations are comparable to other urban/suburban regions (0.45-1.04 ppb). 2. Direct emissions from combustion sources explain nearly 25% of nocturnal HONO concentrations. The authors determined this by examining fresh plumes. 3. Nocturnal HONO formation is RH-dependent and largely explained by heterogeneous surface chemistry. 4. A missing diurnal HONO formation mechanism is a significant source of HONO around noon (average 1.13 ppb/hr).

[Figure]

There are not many long-term records of ambient HONO measurements, and this manuscript provides a valuable dataset to the scientific community. It is well within the scope of ACP and will likely be of interest to ACP readers. I recommend publication after the authors address the following comments.

-Figure 6 shows the HONO/NO2 ratio as a function of RH. The authors state that Fig 6a represents measurements when available surface area is dominated by the ground (i.e., relatively low surface area contributions from aerosols). Can the authors quantify the relative contributions to total surface area from the ground and aerosols? What percentage of the total surface area does the ground represent in clean air and polluted air?

-As shown in Figure 6, HONO/NO2 ratios in polluted air do not decline at RH between 75-95% as is seen in clean air. The authors should provide some explanation here. Why is there a different RH dependency under high PM2.5 conditions?

-The authors claim that the unknown daytime HONO production is different from the heterogeneous nocturnal production (section 3.4). It is not immediately clear how the authors reach this conclusion. They should expand on this statement and provide clear justification.

-A major justification for assuming an unknown HONO source is that the HONO/NO2 ratio rises around noon at peak solar radiation. I have two problems with this that the authors should address. First, any ratio with NO2 in the denominator will increase as NO2 is photolyzed at greater rates. Second – and this is the more serious concern – is that 3-D air quality models predict an increase in HONO/NO2 ratios in the late morning through noon, but they certainly aren't influenced by missing HONO sources (e.g. Figure 8 in http://dx.doi.org/10.1016/j.atmosenv.2015.04.048). While there may well be a significant unknown HONO source during the day, relying on HONO/NO2 ratios does not sufficiently make the case.

-Assuming the existence of a missing HONO source during the day, to what extent

could it be explained by soil emissions?

-The authors state that mass concentration of PM2.5 is likely not the only factor affecting HONO formation on aerosol surfaces. This makes sense intuitively. Do the authors have speciated PM2.5 measurements during this time? How does the chemical composition of aerosols change throughout the year? Would these changes make the NO2 to HONO conversion more or less likely?

Other minor comments: -Check the in-text references to Figures and Tables. Some of the Figures are mis-referenced (e.g. referencing Fig 5 when, in fact, the figure being referenced is Fig 6). This happens quite often in the latter half of the manuscript. -The last sentence in the second paragraph of section 3.3.2 is particularly confusing. -To improve readability, try to have a native English speaker proofread the manuscript. Some of the phrases are oddly worded and obscure the authors' meaning.

---

## Author Comment (AC1) · 2 Aug 2019

**Referee #1**

This paper reports year round measurements of HONO in Nanjing , Eastern China. The effect of direct emissions of HONO is calculated by looking at fresh plumes and production of HONO from heterogeneous reaction of NO2 on aerosol surfaces was speculated upon using nighttime HONO and RH measurements. A HONO budget was calculated, along with a missing HONO source. The effect of the measured HONO on the OH budget was also described. Understanding the role of HONO is crucial for the understanding of oxidation chemistry, especially in the urban environment therefore this study is important work that should be published. There are very few long term measurements of HONO in the literature, with most studies being done in short term campaigns. The analysis here is a reasonable attempt at understanding the role of HONO, albeit with a fairly limited set of supporting measurements. It is within scope of ACP and I recommend publication subject to completion of the following modifications.

*General comments:*
*-No OH measurements were available during the measurements period so the authors have calculated OH concentrations for their analysis (P7). They use the work of Rohrer and Berresheim that correlates OH with J(O1D). I find this a strange choice of literature to use as it was based on work in a very different environment. I believe there are numerous measurements of OH taken within the PRD that would be a more relevant way to infer OH concentrations for this study.*
Response: Thanks for the comment and suggestion. We have reset the parameters a, b, and c, based on the OH studies in the PRD, China.

*Line 190-201: The coefficient a reflects the general chemical conditions (e.g. $NO_x$ or VOCs) at the selected place for research, and the exponent b represents the combined effects of all photolytic processes on OH, and the parameter c counts the light-independent OH sources. The values of a and b in Eq. (2) are adopted from the study in the Pearl River Delta (Lu et al., 2012). The value of c is set to $1.0\times10^6$ $cm^{-3}$, a typical nighttime OH concentration in urban areas of China (Li et al., 2012;Lu et al., 2014). The calculated OH concentrations around noon were in the range of $0.15$-$1.6\times10^7$ $cm^{-3}$, comparable to observations in Chinese urban atmospheres (Lu et al., 2012;Lu et al., 2013).*

$$[OH] = a \times (J(O^1D)/10^{-5}\,s^{-1})^b + c$$
$$(a = 5.6\times10^6\,cm^{-3},\ b = 0.68,\ c = 1.0\times10^6\,cm^{-3})$$

(2)

*-The authors present a calculation of the effect of HONO on OH formation and compare it to formation by O3 photolysis. This study needs expanding a little. OH production from the HO2 + NO reaction would likely be the largest source in such an environment as this study. If the authors want to look at HOx radical formation then they should also make some comment about the effect of other sources such as HCHO photolysis to form HO2 and O3 + alkene reactions. I realise they may not have the supporting measurements to do this accurately but some mention should be made of it.*

Response: Thanks for the comment and suggestion. We have added the content about the sources of OH radicals in the revised manuscript.

*Line 279-285: In addition to the two mechanisms mentioned above, there are other pathways to generate primary OH radicals: the photolysis of aldehydes, especially HCHO, can form $HO_2$ radicals, and then react with NO to form OH radicals; the reaction of ozone with alkenes produce OH radicals directly; the ozonolysis of alkenes and nighttime reactions of $NO_3$ radicals with alkenes can also be net sources of OH radicals (Finlayson-Pitts and Pitts, 2000;Seinfeld and Pandis, 2016).*

*-Some comment should be made as to how much the 'missing HONO' source contributes to OH. This is important in terms of understanding how much models might be underestimating OH by not having all the HONO sources in them.*

Response: Thanks for the comment and suggestion. We have quantified the relative contribution of the missing HONO source to OH in the revised manuscript.

*Line 549-552: In our study, the OH production rate from the missing HONO accounts for about 78% of total $P_{OH}(HONO)$ (Fig. S2), suggesting that the unconventional source of HONO is of significance to atmospheric oxidation.*

[Figure]

Average OH production rates from photolysis of HONO, the missing HONO and $O_3$ around noon (10:00-14:00 LT), from Nov. 2017 to Nov. 2018.

*-Following on from this, some mention should be made about how various air quliaty forecasting or regional models treat HOHO and how adding in the 'unconventional' sources might affect oxidation chemistry.*

Response: Thanks for the comment and suggestion. We have recommended a possible mechanism that can be adopted into the models in the in the revised manuscript, i.e. the photo-induced heterogeneous reaction of $NO_2$.

*Line 572-577: Our study suggest that the missing source of HONO should be considered in the air quality forecasting or regional models to characterize atmospheric oxidizing capacity better, especially in warm seasons (spring and summer). Based on the measurement (Fig. S3), the light-induced heterogeneous conversion of $NO_2$ to HONO on aerosol surfaces and ground surface can been included in simulation works probably, as what did in the study of Lee et al. (2016).*

**Minor comments:**

*-P5 line 150: Can the authors justify that the measurement is 'interference free'? It was my understanding that the LOPAP instrument is subject to interference from other nitrate species. Please expand this.*

Response: Thanks for the comment and suggestion. The interferences can be reduced mostly but possibly not completely by subtracting the signal of channel 2 from the signal of channel 1, so we have modified the statement in the revised manuscript

*Line 143-146: In the first stripping coil, all of the HONO and a fraction of interfering substances were absorbed into solution, and the remaining interfering species ($NO_2$, $HNO_3$, PAN, etc.) were absorbed in the second stripping coil.*

*Line 149-151: The real HONO signal was the difference between the signals in the two channels, and the interferences can be minimized by this method.*

*P8 line 244: 'in the' should be 'at'*

Response: Thanks for the suggestion. We have corrected this in the revised manuscript.

*Line 248-251: Given that the photolytic lifetime of HONO is about 10-20 min at the midday (Stutz et al., 2000), the considerable HONO concentration during daytime indicates the existence of strong production of   HONO.*

*P9 Line 264: The authors describe HONO as an efficient reservoir of OH radicals. I'm not totally convinced this is the correct way to describe it. A reservoir suggests a long lived species that enables transport of OH radicals. I would have thought the lifetime of HONO would be very short, maybe the authors could comment on this.*

Response: Thanks for the comment and suggestion. We have modified the statements in the revised manuscript.

*Line 14-16: Nitrous acid (HONO), a important precursor of the hydroxyl radical (OH), has been long-standing recognized to be of significance to atmospheric chemistry, but its sources are still debate.*

*Line 269-270: The elevated mixing ratio of HONO presents an efficient source of OH radicals during daytime in Nanjing.*

**Reference**

Finlayson-Pitts, B. J., and Pitts, J. N.: CHAPTER 6 - Rates and Mechanisms of Gas-Phase Reactions in Irradiated Organic – NOx – Air Mixtures, in: Chemistry of the Upper and Lower Atmosphere, edited by: Finlayson-Pitts, B. J., and Pitts, J. N., Academic Press, San Diego, 179-263, 2000.

Lee, J. D., Whalley, L. K., Heard, D. E., Stone, D., Dunmore, R. E., Hamilton, J. F., Young, D. E., Allan, J. D., Laufs, S., and Kleffmann, J.: Detailed budget analysis of HONO in central London reveals a missing daytime source, Atmospheric Chemistry and Physics, 16, 2747-2764, 10.5194/acp-16-2747-2016, 2016.

Li, X., Brauers, T., Häseler, R., Bohn, B., Fuchs, H., Hofzumahaus, A., Holland, F., Lou, S., Lu, K. D., Rohrer, F., Hu, M., Zeng, L. M., Zhang, Y. H., Garland, R. M., Su, H., Nowak, A., Wiedensohler, A., Takegawa, N., Shao, M., and Wahner, A.: Exploring the atmospheric chemistry of nitrous acid (HONO) at a rural site in Southern China, Atmospheric Chemistry and Physics, 12, 1497-1513, 10.5194/acp-12-1497-2012, 2012.

Lu, K. D., Rohrer, F., Holland, F., Fuchs, H., Bohn, B., Brauers, T., Chang, C. C., Haseler, R., Hu, M., Kita, K., Kondo, Y., Li, X., Lou, S. R., Nehr, S., Shao, M., Zeng, L. M., Wahner, A., Zhang, Y. H., and Hofzumahaus, A.: Observation and modelling of OH and HO2 concentrations in the Pearl River Delta 2006: a missing OH source in a VOC rich atmosphere, Atmospheric Chemistry and Physics, 12, 1541-1569, 10.5194/acp-12-1541-2012, 2012.

Lu, K. D., Hofzumahaus, A., Holland, F., Bohn, B., Brauers, T., Fuchs, H., Hu, M., Haseler, R., Kita, K., Kondo, Y., Li, X., Lou, S. R., Oebel, A., Shao, M., Zeng, L. M., Wahner, A., Zhu, T., Zhang, Y. H., and Rohrer, F.: Missing OH source in a suburban environment near Beijing: observed and modelled OH and HO2 concentrations in summer 2006, Atmospheric Chemistry and Physics, 13, 1057-1080, 10.5194/acp-13-1057-2013, 2013.

Lu, K. D., Rohrer, F., Holland, F., Fuchs, H., Brauers, T., Oebel, A., Dlugi, R., Hu, M., Li, X., Lou, S. R., Shao, M., Zhu, T., Wahner, A., Zhang, Y. H., and Hofzumahaus, A.: Nighttime observation and chemistry of HOx in the Pearl River Delta and Beijing in summer 2006, Atmospheric Chemistry and Physics, 14, 4979-4999, 10.5194/acp-14-4979-2014, 2014.

Seinfeld, J. H., and Pandis, S. N.: Atmospheric chemistry and physics: from air pollution to climate change, John Wiley & Sons, 2016.

Stutz, J., Kim, E. S., Platt, U., Bruno, P., Perrino, C., and Febo, A.: UV-visible absorption cross sections of nitrous acid, Journal of Geophysical Research: Atmospheres, 105, 14585-14592, 10.1029/2000jd900003, 2000.

---

## Author Comment (AC2) · 2 Aug 2019

**Referee #3**

This manuscript describes continuous ambient measurements of HONO, NO, NO2, and PM2.5 at the SORPES station in Nanjing (eastern China) from November 2017 to November 2018. The main conclusions are:

1. Seasonal average HONO concentrations are comparable to other urban/suburban regions (0.45-1.04 ppb). 2. Direct emissions from combustion sources explain nearly 25% of nocturnal HONO concentrations. The authors determined this by examining fresh plumes. 3. Nocturnal HONO formation is RH-dependent and largely explained by heterogeneous surface chemistry. 4. A missing diurnal HONO formation mechanism is a significant source of HONO around noon (average 1.13 ppb/hr).

There are not many long-term records of ambient HONO measurements, and this manuscript provides a valuable dataset to the scientific community. It is well within the scope of ACP and will likely be of interest to ACP readers. I recommend publication after the authors address the following comments.

*-Figure 6 shows the HONO/NO2 ratio as a function of RH. The authors state that Fig 6a represents measurements when available surface area is dominated by the ground (i.e., relatively low surface area contributions from aerosols). Can the authors quantify the relative contributions to total surface area from the ground and aerosols? What percentage of the total surface area does the ground represent in clean air and polluted air?*

Response: Thanks for the comment and suggestion.

We calculated aerosol surface density from the particle number size distributions between 6 nm and 820 nm, by assuming that all particles are spherically shaped. and We calculated ground surface density through the equation: $(\frac{S}{V})_{grd} = \frac{1}{H}$, where H is the height of boundary layer, and a value of 100m is assumed for nighttime (Su et al., 2008). As the following figure shows, the surface area to volume ratio of ground is dominant, but under the condition of severe pollution, the aerosol can contribute about 10% of the total surface area. Besides the surface area, the conversion of $NO_2$ to HONO should also be determined by the surface reactivity, i.e. the uptake coefficient of $NO_2$-to-HONO ($\gamma_{NO_2 \rightarrow HONO}$). Differ from the prolonged exposure to oxidizing agents and radiation of the ground surface, the aerosol surface is relatively more fresh, and possibly more reactive. For example, the reduction of $NO_2$ in the presence of water by C–O and C–H groups in the soot is proposed to produce HONO quickly (Ammann et al., 1998). In our study, in case we assume that all of the observed HONO is formed on particle surfaces at night, the derived $\gamma_{NO_2 \rightarrow HONO}$ is $1.44 \times 10^{-5}$, within a reasonable range of laboratory measurements.

[Figure]

the averaged surface area to volume ratio ($m^{-1}$) of ground and aerosol in clean air ($PM_{2.5}<25\mu g/m^3$) and polluted air ($PM_{2.5}>150\mu g/m^3$)

*Line 481-487: For 30%-100% of the measured mean $C_{HONO}$ (0.0043 $h^{-1}$) in winter, the uptake coefficient of $NO_2$-to-HONO ($\gamma_{NO_2 \rightarrow HONO}$) calculated from Eq. (8) is in the range of $6.9 \times 10^{-6}$ to $1.44 \times 10^{-5}$, consistent with the results from many laboratory studies which demonstrate that the uptake coefficients of $NO_2$ ($\gamma_{NO_2}$) on multiple aerosol surfaces or wet surfaces are mainly distributed around $10^{-5}$ with the HONO yield varying from 0.1 to 0.9 (Grassian, 2002;Aubin and Abbatt, 2007;Khalizov et al., 2010;Han et al., 2017).*

**-As shown in Figure 6, HONO/NO2 ratios in polluted air do not decline at RH between 75-95% as is seen in clean air. The authors should provide some explanation here. Why is there a different RH dependency under high PM2.5 conditions?**
Response: Thanks for the comment and suggestion. We have added some discussion into the revised manuscript.

*Line 427-438: With the increase of RH, the hygroscopic growth of aerosol particles should provide larger surface area. When RH is higher than 75%, which has exceeded the mutual deliquescence relative humidity of inorganic salts (Fountoukis and Nenes, 2007), aerosols will transfer to aqueous phase gradually, and then promoting multiphase or heterogeneous chemistry processes (Herrmann et al., 2015). For example, the oxidation of $SO_2$ by $NO_2$ on aqueous aerosol surface may produce $NO_2^-$/HONO efficiently under polluted condition (Xie et al., 2015;Wang et al., 2016). In addtion, the enhancement $NO_2$ uptake on micro-droplets by anions has been reported in experiments (Yabushita et al., 2009).*

*-The authors claim that the unknown daytime HONO production is different from the heterogeneous nocturnal production (section 3.4). It is not immediately clear how the authors reach this conclusion. They should expand on this statement and provide clear justification.*

Response: Thanks for the comment and suggestion. The highest noontime $P_{unknown}$ value is 1.72 ppb/h in spring, followed by 1.11 ppb/h in summer, 0.66 ppb/h in autumn and 0.58 ppb/h in autumn, unlike the seasonal variation of $NO_2$; and $P_{unknown}$ shows an increase towards noon, which is also distinguished from the diurnal pattern of $NO_2$. These results indicate that there must be some other factors affecting $P_{unknown}$, in case $NO_2$ is assumed to be a dominate precursor of HONO at daytime

*Line 558-561: The average value of $P_{unknown}$ normalized by $NO_2$ is 0.1 $h^{-1}$, over 18 times greater than the nighttime conversion rate (0.0055 $h^{-1}$), also implying that $P_{unknown}$ cannot be explained by the nocturnal mechanism of $NO_2$-to-HONO.*

*-A major justification for assuming an unknown HONO source is that the HONO/NO2 ratio rises around noon at peak solar radiation. I have two problems with this that the authors should address. First, any ratio with NO2 in the denominator will increase as NO2 is photolyzed at greater rates. Second – and this is the more serious concern – is that 3-D air quality models predict an increase in HONO/NO2 ratios in the late morning through noon, but they certainly aren't influenced by missing HONO sources (e.g. Figure 8 in http://dx.doi.org/10.1016/j.atmosenv.2015.04.048). While there may well be a significant unknown HONO source during the day, relying on HONO/NO2 ratios does not sufficiently make the case.*

Response: Thanks for the comments.

For first problem, we agree that the greater rates of $NO_2$ can also increase the $HONO/NO_2$ ratio. If just considering of the photolysis of HONO and $NO_2$, both of which will convert to NO, the loss of HONO and the almost unchanged concentration of $NO_x$ ($NO_2$+NO) will reduce the ratio $HONO/NO_x$. So we actually use the ratio $HONO/NO_x$ to present the conversion of $NO_x$ to HONO partly (please see Fig.1 and Fig. 2 in the revised manuscript).

For second problem, the increase of $HONO/NO_x$ at daytime can result from: (1) the homogeneous reaction of NO and OH radical (R3); (2) the conversion of $NO_2$ to HONO (R4, R5); (3) other $NO_x$-independent sources. In the work of Couzo et al. (2015) (Figure 8 in http://dx.doi.org/10.1016/j.atmosenv.2015.04.048), when they only considered R3, the predicted daytime $HONO/NO_2$ can follow the time variation of the measured ratio but underestimate significantly, and after include the heterogeneous formation from $NO_2$ (R4, R5) and $HNO_3$ (R6), the simulated $HONO/NO_2$ was improved during daytime, but significantly contradicted with the observed value in the second half of the night. Until now, the heterogeneous reaction mechanisms (R4, R5, R6) are actually not clear yet, there are uncertainties involved with the parameterizations in various models, many simulation works still tend to underestimate HONO concentrations (Czader et al., 2012;Lee et al., 2016).

The missing source ($P_{unknown}$) defined in our study contains the heterogeneous processes mentioned above. We want to understand which mechanism might be more important based on our measurements. The source of HONO is divided into gas phase reaction (R3), combustion emission and unknown source $P_{unknown}$. So both the homogeneous formation and unknown source of HONO can increase the HONO/$NO_x$ ratio at daytime, with a mean value of 0.71 ppb/h and 1.02 ppb/h, respectively. $P_{unknown}$ has found to correlated with $NO_2$*UVB, indicating the photo-induced heterogeneous conversion of $NO_2$ to HONO, but for now we do not have any solid evidence to identify which surface (ground surface and aerosol surface) are important in this potential mechanism.

*Line 534:*

$$P_{unknown} = J(HONO)[HONO] + k_{HONO+OH}[HONO][OH] + \frac{\nu_{HONO}}{H}[HONO]$$
$$+ \frac{\Delta HONO}{\Delta t} - k_{NO+OH}[NO][OH] - \frac{0.79\% \times \Delta NO_x}{\Delta t} \tag{10}$$

*Line 261-264: If the HONO sources during daytime are consistent with those at night, the minimum HONO/$NO_x$ ratios should occur at noon due to the intense photochemical loss of HONO. Therefore, there must be additional sources of HONO during daytime (e.g. R3).*

*Line 539-542: the average homogeneous reaction rate between NO and OH ($P_{NO+OH}$) is 0.71 ppb/h and $P_{emis}$ just gives a tiny part of HONO at a rate of 0.02 ppb/h, meaning that most of HONO comes from an unknown source whose average rate ($P_{unknown}$) is 1.02 ppb/h, contributing about 58% of the production of HONO.*

**-Assuming the existence of a missing HONO source during the day, to what extent could it be explained by soil emissions?**
Response: Thanks for the comment and suggestion.

The averaged missing source calculated in our study is 1.02 ppb/h around noon (10:00-14:00 LT). So far, we cannot exclude the potential contribution from (photo-enhanced) heterogeneous reaction of $NO_2$, and the photolysis of adsorbed nitric acid ($HNO_3$) and particulate nitrate ($NO_3^-$). It's difficult to derive the rate or the amount of HONO emitted from soil emission, the main reason is that we were lack of direct observation. However, we are still trying to estimate the contribution of soil emissions to HONO through solving overdetermined equations at night, due to the relatively simple sources of HONO and without the influences of HONO photolysis, and the mixing effect of boundary layer (see part 4 in the revised manuscript for details). And, in average, 14.5% of nighttime HONO is found to be explained by soil emissions. The key to our calculation is the assumption that the mixing level of observed $NH_3$ can represent the intensity of soil emission of HONO. Although the processes of HONO and $NH_3$ emission from soil may not be completely synchronized, the seasonal patterns for each should be consistent.

*Line 602-614: Although we do not directly measure HONO emissions from soil, the observed ammonia can represent its monthly average intensity, based on the following hypothesis: the dominant source of NH₃ is from soil, especially from fertilizers (NH₄+→NH₃) for a good correlation between ammonia and temperature in the site (r=0.63, p=0.01), omitting the contributions of livestock to NH₃ since there is only a small poultry facility within 10 km of this site (Meng et al., 2011;Huang et al., 2012;Behera et al., 2013). Combustion sources (vehicles, industry, biomass burning) should contribute only a fraction of NH₃ seeing that NH₃ is not related to NOx or CO in our study. Moreover, the release of both HONO and NH₃ depend on the strength of microbial activities, fertilizing amount, and soil properties (e.g., temperature, acidity and water content of soil). Although the processes of HONO and NH₃ emission from soil may not be completely synchronized, the seasonal patterns for each should be consistent.*

**-The authors state that mass concentration of PM2.5 is likely not the only factor affecting HONO formation on aerosol surfaces. This makes sense intuitively. Do the authors have speciated PM2.5 measurements during this time? How does the chemical composition of aerosols change throughout the year? Would these changes make the NO2 to HONO conversion more or less likely?**

Response: Thanks for the comment and suggestion.

The seasonal variation of aerosol compositions has been reported in our previously work, showed in the following first figure: the particulate nitrate exhibits a maximum value in January and a minimum in August, and particulate sulfate shows a relatively weak seasonal cycle (Sun et al., 2018). An intuitive conclusion is that the proportion of nitrate will increase and the proportion of sulfate will decrease with the aerosol loading, from summer to winter.

[Figure]

Monthly averaged nitrate (blue), sulfate (red), $NO_x$ (orange) mass concentrations and nitrate to sulfate molar-based ratio (grey) measured at the SORPES station during March 2014 to February 2016 (Sun et al., 2018).

The slope of $HONO_{corr}/NO_2$ and $PM_{2.5}$ varies over a relatively wide range, caused by some unknown factors that need to be explored. As the following figure shows, when the proportion of nitrate in aerosol is higher, the slope of $HONO_{corr}/NO_2$ and $PM_{2.5}$ tend to be lower slightly while the relationship shows differently for sulfate. The value of $(PM_{2.5}-NO_3^--SO_4^{2-}-NH_4^+)/PM_{2.5}$ can roughly represent the ratio of organic compounds in most situations, and it seems that the high ratio of organic aerosol occurs with the high slope of $HONO_{corr}/NO_2$ and $PM_{2.5}$. But simply relying on these cannot make too much sense, for example, the heat can make particulate nitrate volatilize into nitric acid gas and cause soil to emit more HONO, so we can see the highest $HONO_{corr}/NO_2$ ratio and the lowest proportion of nitrate to aerosol in summer. In future work, we're going to study the impact of aerosol components to the heterogeneous formation of HONO through laboratory experiments.

[Figure]

Scatter plot of HONO$_{corr}$/NO$_2$ and PM$_{2.5}$ in the time (3:00-6:00 LT) when HONO$_{corr}$/NO$_2$ reaches the pseudo steady state each night and are colored by the ratios of main aerosol compositions: (a) NO$_3^-$/PM$_{2.5}$, (b) SO$_4^{2-}$/PM$_{2.5}$, (c) NH$_4^+$/PM$_{2.5}$, (d) others/PM$_{2.5}$, i.e. (PM$_{2.5}$-NO$_3^-$-SO$_4^{2-}$-NH$_4^+$)/PM$_{2.5}$.

*Other minor comments:*
*-Check the in-text references to Figures and Tables. Some of the Figures are mis-referenced (e.g. referencing Fig 5 when, in fact, the figure being referenced is Fig 6). This happens quite often in the latter half of the manuscript.*
Response: Thanks. We have re-edited the references to Figures and Tables in the revised manuscript.

*- The last sentence in the second paragraph of section 3.3.2 is particularly confusing.*
Response: Thanks. We have re-edited the language in the revised manuscript.

*Line 393-398: Even at the lowest measured RH of 18%, the absolute moisture content in the atmosphere is still greater than $10^3$ ppm in our study, which is quite abundant to react with*

*NO₂, but the HONO$_{corr}$/NO₂ ratio is quite small and remains unchanged when RH is below 45%, indicating that the NO₂ to HONO conversion efficiency should be determined by water covering the surfaces, rather than by the amount of water in the air.*

**-To improve readability, try to have a native English speaker proofread the manuscript. Some of the phrases are oddly worded and obscure the authors' meaning.**
Response: Thanks for the suggestion.

**Reference**

Ammann, M., Kalberer, M., Jost, D., Tobler, L., Rossler, E., Piguet, D., Gaggeler, H., and Baltensperger, U.: Heterogeneous production of nitrous acid on soot in polluted airmasses, NATURE, 395, 157-160, 10.1038/25965, 1998.

Aubin, D. G., and Abbatt, J. P.: Interaction of NO2 with hydrocarbon soot: Focus on HONO yield, surface modification, and mechanism, The Journal of Physical Chemistry A, 111, 6263-6273, 2007.

Behera, S. N., Sharma, M., Aneja, V. P., and Balasubramanian, R.: Ammonia in the atmosphere: a review on emission sources, atmospheric chemistry and deposition on terrestrial bodies, Environ Sci Pollut Res Int, 20, 8092-8131, 10.1007/s11356-013-2051-9, 2013.

Couzo, E., Lefer, B., Stutz, J., Yarwood, G., Karamchandani, P., Henderson, B., and Vizuete, W.: Impacts of heterogeneous HONO formation on radical sources and ozone chemistry in Houston, Texas, Atmos. Environ., 112, 344-355, 10.1016/j.atmosenv.2015.04.048, 2015.

Czader, B. H., Rappenglück, B., Percell, P., Byun, D. W., Ngan, F., and Kim, S.: Modeling nitrous acid and its impact on ozone and hydroxyl radical during the Texas Air Quality Study 2006, Atmospheric Chemistry and Physics, 12, 6939-6951, 10.5194/acp-12-6939-2012, 2012.

Fountoukis, C., and Nenes, A.: ISORROPIA II: a computationally efficient thermodynamic equilibrium model for K+-Ca2+-Mg2+-Nh(4)(+)-Na+-SO42--NO3--Cl--H2O aerosols, Atmospheric Chemistry and Physics, 7, 4639-4659, 10.5194/acp-7-4639-2007, 2007.

Grassian, V.: Chemical reactions of nitrogen oxides on the surface of oxide, carbonate, soot, and mineral dust particles: Implications for the chemical balance of the troposphere, The Journal of Physical Chemistry A, 106, 860-877, 2002.

Han, C., Liu, Y., and He, H.: Heterogeneous reaction of NO2 with soot at different relative humidity, Environmental Science and Pollution Research, 24, 21248-21255, 10.1007/s11356-017-9766-y, 2017.

Herrmann, H., Schaefer, T., Tilgner, A., Styler, S. A., Weller, C., Teich, M., and Otto, T.: Tropospheric aqueous-phase chemistry: kinetics, mechanisms, and its coupling to a changing gas phase, Chem Rev, 115, 4259-4334, 10.1021/cr500447k, 2015.

Huang, X., Song, Y., Li, M., Li, J., Huo, Q., Cai, X., Zhu, T., Hu, M., and Zhang, H.: A high-resolution ammonia emission inventory in China, Global Biogeochemical Cycles, 26, n/a-n/a, 10.1029/2011gb004161, 2012.

Khalizov, A. F., Cruz-Quinones, M., and Zhang, R.: Heterogeneous reaction of NO2 on fresh and coated soot surfaces, The Journal of Physical Chemistry A, 114, 7516-7524, 2010.

Lee, J. D., Whalley, L. K., Heard, D. E., Stone, D., Dunmore, R. E., Hamilton, J. F., Young, D. E., Allan, J. D., Laufs, S., and Kleffmann, J.: Detailed budget analysis of HONO in central London reveals a missing daytime source, Atmospheric Chemistry and Physics, 16, 2747-2764, 10.5194/acp-16-2747-2016, 2016.

Meng, Z., Lin, W., Jiang, X., Yan, P., Wang, Y., Zhang, Y., Jia, X., and Yu, X.: Characteristics of atmospheric ammonia over Beijing, China, Atmospheric Chemistry and Physics, 11, 6139-6151, 2011.

Su, H., Cheng, Y. F., Cheng, P., Zhang, Y. H., Dong, S., Zeng, L. M., Wang, X., Slanina, J., Shao, M., and Wiedensohler, A.: Observation of nighttime nitrous acid (HONO) formation at a non-urban site during PRIDE-PRD2004 in China, Atmos. Environ., 42, 6219-6232, 10.1016/j.atmosenv.2008.04.006, 2008.

Sun, P., Nie, W., Chi, X., Xie, Y., Huang, X., Xu, Z., Qi, X., Xu, Z., Wang, L., Wang, T., Zhang, Q., and Ding, A.: Two years of online measurement of fine particulate nitrate in the western Yangtze River Delta: influences of thermodynamics and $N_2O_5$ hydrolysis, Atmospheric Chemistry and Physics, 18, 17177-17190, 10.5194/acp-18-17177-2018, 2018.

Wang, G., Zhang, R., Gomez, M. E., Yang, L., Levy Zamora, M., Hu, M., Lin, Y., Peng, J., Guo, S., Meng, J., Li, J., Cheng, C., Hu, T., Ren, Y., Wang, Y., Gao, J., Cao, J., An, Z., Zhou, W., Li, G., Wang, J., Tian, P., Marrero-Ortiz, W., Secrest, J., Du, Z., Zheng, J., Shang, D., Zeng, L., Shao, M., Wang, W., Huang, Y., Wang, Y., Zhu, Y., Li, Y., Hu, J., Pan, B., Cai, L., Cheng, Y., Ji, Y., Zhang, F., Rosenfeld, D., Liss, P. S., Duce, R. A., Kolb, C. E., and Molina, M. J.: Persistent sulfate formation from London Fog to Chinese haze, Proc Natl Acad Sci U S A, 113, 13630-13635, 10.1073/pnas.1616540113, 2016.

Xie, Y., Ding, A., Nie, W., Mao, H., Qi, X., Huang, X., Xu, Z., Kerminen, V.-M., Petäjä, T., Chi, X., Virkkula, A., Boy, M., Xue, L., Guo, J., Sun, J., Yang, X., Kulmala, M., and Fu, C.: Enhanced sulfate formation by nitrogen dioxide: Implications from in situ observations at the SORPES station, Journal of Geophysical Research: Atmospheres, 120, 12679-12694, 10.1002/2015jd023607, 2015.

Yabushita, A., Enami, S., Sakamoto, Y., Kawasaki, M., Hoffmann, M. R., and Colussi, A. J.: Anion-Catalyzed Dissolution of NO2 on Aqueous Microdroplets, J. Phys. Chem. A, 113, 4844-4848, 10.1021/jp900685f, 2009.

---

## Editor Comment (EC1) · Jianzhong Ma (Editor) · 4 Aug 2019

Regarding to the calculation of [OH] using Eq. (2), I doubt if the concern from Referee #1 has been well addressed. The assumption that [OH] changes as a function of unique parameter J(O1D) with other influencing factors being constant and the same for different regions needs to be reconsidered carefully.

---

## Editor Decision (ED1)

Dear Dr. Nie,

I sent the revised version of your manuscript (Liu et al., 2019) to two experts (Referee #1 and Referee #3) for further review. Now I have received a report from Referee #3. Unfortunately, Referee #1 declined my invitation. In the first round of review, the major concerns from Referee #3 are about the calculation of OH concentration and the use of chemical reactions for OH production. I agree with the referee. Moreover, I think using simple parametrization of OH by $J(O^1D)$ does not fit and should be discouraged in this and other similar studies.

Equation (2) in the original version of the manuscript (referred to as E2_ori hereafter) was derived by Rohrer and Berresheim (2006) based on measurements at a GAW site in southern Germany. There is no doubt that such a strong correlation between OH and $J(O^1D)$ exists in the clean area (where $NO_x$ levels are low). It has been shown that the reaction $O(^1D)+H_2O$ dominates the global mean tropospheric source of OH (Lelieveld et al., 2016). However, E2_ori might not work well for the polluted area (where $NO_x$ levels are high).

Equation (2) in the revised version of the manuscript (referred to as E2_rev hereafter) was derived by Lu et al. (2012) based on measurements at a site in the Pearl River Delta. Lu et al. (2012) performed the fittings of measurement data without considering the variability of $NO_x$ and VOCs. E2_rev is an updated version of E2_ori with only a, b and c values changed, while the expression is functionally the same as that of E2_ori. It is doubtful if E2_rev can also be used to the Yangtze Delta. Actually, different a, b and c values were derived by fitting the OH and $J(O^1D)$ observed in urban Beijing (Lu et al., 2013).

Note that both E2_ori and E2_rev were used to investigate the correlation between observed OH concentration and $J(O^1D)$. There might be large biases in calculated OH from this parameterization especially under high NOx conditions, even for the same site where the measurements were carried out (e.g., E2_rev for the PRD site (Lu et al., 2012)). In contrast to the tropical and clean regions, the reaction $HO_2+NO$ (as well as $RO_2+NO$) contributes more to OH production than $O(^1D)+H_2O$ in the polluted areas, including eastern China (e.g., Lu et al., 2012;Ma et al., 2012;Tang et al., 2015;Lu et al., 2017). Variations of NO and VOCs, both of which influence the cycles among radicals (Lelieveld et al., 2008;Hofzumahaus et al., 2009), may have a large effect on the OH levels, which are not taken into account in this study.

This study shows that the photolysis of HONO (J(HONO)) has a larger contribution to primary production of OH than the effective photolysis of ozone ($J(O^1D)$), in consistent with the results from previous studies (e.g., An et al., 2009;Tang et al., 2015). However, I cannot see the role of HONO or J(HONO) in E2_rev. The OH calculated from E2_rev is used in several equations (e.g., Eq. 3). Should the result (e.g., $P_{OH}(HONO)$ in Eq. 3) derived by using OH, which is primarily estimated from

E2_rev, have a strong feedback on the OH value? Moreover, could you explain why the second term (- $k_{NO+OH}$[NO][OH]) should be included on the right side of Eq. 3? I do not understand the meaning of $P_{unknown}$ in Eq. (10), some terms of which seems to be the same as the terms $L_{phot}$ or $L_{HONO+OH}$ in Eq. (9).

It would be very appreciated if these issues could be clarified before the acceptance of your manuscript.

Sincerely,

Jianzhong Ma

**References**

An, J. L., Zhang, W., and Qu, Y.: Impacts of a strong cold front on concentrations of HONO, HCHO, $O_3$, and $NO_2$ in the heavy traffic urban area of Beijing, Atmos. Environ., 43, 3454-3459, 10.1016/j.atmosenv.2009.04.052, 2009.

Hofzumahaus, A., Rohrer, F., Lu, K., Bohn, B., Brauers, T., Chang, C.-C., Fuchs, H., Holland, F., Kita, K., Kondo, Y., Li, X., Lou, S., Shao, M., Zeng, L., Wahner, A., and Zhang, Y.: Amplified trace gas removal in the troposphere, Science, 324, 1702-1704, 10.1126/science.1164566, 2009.

Lelieveld, J., Butler, T. M., Crowley, J. N., Dillon, T. J., Fischer, H., Ganzeveld, L., Harder, H., Lawrence, M. G., Martinez, M., Taraborrelli, D., and Williams, J.: Atmospheric oxidation capacity sustained by a tropical forest, Nature, 452, 737-740, 2008.

Lelieveld, J., Gromov, S., Pozzer, A., and Taraborrelli, D.: Global tropospheric hydroxyl distribution, budget and reactivity, Atmos. Chem. Phys., 16, 12477-12493, 10.5194/acp-16-12477-2016, 2016.

Liu, Y., Nie, W., Xu, Z., Wang, T., Wang, R., Li, Y., Wang, L., Chi, X., and Ding, A.: Contributions of different sources to nitrous acid (HONO) at the SORPES station in eastern China: results from one-year continuous observation, Atmos. Chem. Phys. Discuss., 2019, 1-47, 10.5194/acp-2019-219, 2019.

Lu, K. D., Rohrer, F., Holland, F., Fuchs, H., Bohn, B., Brauers, T., Chang, C. C., Häseler, R., Hu, M., Kita, K., Kondo, Y., Li, X., Lou, S. R., Nehr, S., Shao, M., Zeng, L. M., Wahner, A., Zhang, Y. H., and Hofzumahaus, A.: Observation and modelling of OH and HO2 concentrations in the Pearl River Delta 2006: a missing OH source in a VOC rich atmosphere, Atmos. Chem. Phys., 12, 1541-1569, 10.5194/acp-12-1541-2012, 2012.

Lu, K. D., Hofzumahaus, A., Holland, F., Bohn, B., Brauers, T., Fuchs, H., Hu, M., Häseler, R., Kita, K., Kondo, Y., Li, X., Lou, S. R., Oebel, A., Shao, M., Zeng, L. M., Wahner, A., Zhu, T., Zhang, Y. H., and Rohrer, F.: Missing OH source in a suburban environment near Beijing: observed and modelled OH and HO2 concentrations in summer 2006, Atmos. Chem. Phys., 13, 1057-1080, 10.5194/acp-13-1057-2013, 2013.

Lu, X., Chen, N., Wang, Y., Cao, W., Zhu, B., Yao, T., Fung, J. C. H., and Lau, A. K. H.: Radical budget and ozone chemistry during autumn in the atmosphere of an urban site in central China, Journal of Geophysical Research: Atmospheres, 122, 3672-3685, 10.1002/2016jd025676, 2017.

Ma, J. Z., Wang, W., Chen, Y., Liu, H. J., Yan, P., Ding, G. A., Wang, M. L., Sun, J., and Lelieveld, J.: The

IPAC-NC field campaign: a pollution and oxidization pool in the lower atmosphere over Huabei, China, Atmos. Chem. Phys., 12, 3883-3908, 10.5194/acp-12-3883-2012, 2012.

Rohrer, F., and Berresheim, H.: Strong correlation between levels of tropospheric hydroxyl radicals and solar ultraviolet radiation, Nature, 442, 184-187, http://www.nature.com/nature/journal/v442/n7099/suppinfo/nature04924_S1.html, 2006.

Tang, Y., An, J., Wang, F., Li, Y., Qu, Y., Chen, Y., and Lin, J.: Impacts of an unknown daytime HONO source on the mixing ratio and budget of HONO, and hydroxyl, hydroperoxyl, and organic peroxy radicals, in the coastal regions of China, Atmos. Chem. Phys., 15, 9381-9398, 10.5194/acp-15-9381-2015, 2015.

---

## Author Response (AR2)

**Dear editor,**

Thanks for your attention and careful review to our work. These comments and suggestions have improved our manuscript very much.

The main concern is about the empirical model to calculate levels of OH radicals (Eq.2). The strong nearly linear correlation between OH and $J(O^1D)$ is proposed by Rohrer and Berresheim (2006), based on the observations in a lightly polluted site, where the reaction of $HO_2$ with NO produce most of OH radicals. This relationship has been also verified by the OH campaigns in the polluted areas of China, where the most striking feature is the almost equal slope of the $OH-J(O^1D)$ relation for different locations and seasons (Rohrer et al., 2014;Tan et al., 2018). We can make assumptions that the comprehensive impact of reactants (e.g. $NO_x$ and VOCs) on OH cannot compete with that of UV light to OH, and the chemical environments of OH are similar in polluted area of China. What's more, in the case of low concentration of NO and high value of $J(O^1D)$, the error of $P_{unknown}$ from OH radicals will be restrained significantly (Figure S1(d)). In summary, the empirical model to calculate the concentrations of OH radicals can be reasonable to a certain extent and the error of derived OH radicals has been assessed as not subverting the relative conclusions in this study (Fig. S1(a) and Fig. S1(d)).

For details, we divided this response into 3 part.

*1. Response to the concern about the empirical model to calculate levels of OH radicals.*

Hydroxyl radicals (OH) in the troposphere mainly come from:

**1. primary production**
-The reaction of $O(^1D)$ atoms, formed in the photolysis of ozone at wavelengths below 320 nm, with water;
-The photolysis of aldehydes (mainly HCHO) can form $HO_2$ radicals, which are converted to OH radicals in the presence of NO;
- The photolysis of HONO upon irradiation with UV light;
- The reactions of ozone with alkenes.

**2. secondary production**
- Reactions of $HO_2$ ($RO_2$), which are produced by the ozonolysis of alkenes, nighttime reactions of $NO_3$ radicals with alkenes, and OH reactions with CO and VOCs, with NO can reproduce OH.

[Figure]

*Schematic drawing of HOx chemistry, illustrating established, major chemical pathways in the troposphere. Radical chemistry is initiated by primary radical production (blue arrows), and radical chain reactions (red arrows) cause cycling between OH, $HO_2$, and $RO_2$ (Lu et al., 2012).*

The strong correlation of between levels of tropospheric hydroxyl radicals and solar ultraviolet radiation, found in Rohrer and Berresheim (2006), is not only caused by the reaction of $O(^1D)$ atoms with $H_2O$. Actually, the observation location in that study is lightly polluted, whose mean concentration of $NO_x$ is 2.19 ppb and max concentration of $NO_x$ is 76.3 ppb, and the reaction of $HO_2$ with NO produce most of OH radicals. A generalized reaction scheme for OH photochemistry has been discussed by Rohrer and Berresheim (2006):

$$O_3 + h\upsilon (+H_2O) \rightarrow 2OH \ (\alpha J(O^1D))$$

$$OH + hydrocarbons, \ CO, \ H_2 \rightarrow HO_2 + products(\tau_{HC}^{-1} = 3.3s^{-1})$$

$$HO_2 + NO \rightarrow OH + NO_2 \ (k_{HO_2+NO} = 8.56 \times 10^{-12} \, cm^3 s^{-1})$$

$$OH + NO_2 \rightarrow HNO_3 \ (k_{OH+NO_2} = 11.5 \times 10^{-12} \, cm^3 s^{-1})$$

$$HO_2 + HO_2(+H_2O) \rightarrow H_2O_2 \ (k_{HO_2+HO_2} = 4.5 \times 10^{-12} \, cm^3 s^{-1})$$

$$NO + O_3 \rightarrow NO_2 \ (k_{NO+O_3} = 1.82 \times 10^{-14} \, cm^3 s^{-1})$$

$$NO_2 + h\upsilon \rightarrow NO + O_3 \ (J(NO_2))$$

$$\alpha = \frac{k_{O^1D+H_2O}[H_2O]}{k_{O^1D+H_2O}[H_2O] + k_{O^1D+N_2}[N_2] + k_{O^1D+O_2}[O_2]}$$

By using the steady-state approach for OH, HO$_2$, and NO, and neglecting minor terms, finally deriving the final expression for OH:

$$[OH] = \frac{k_{HO_2+NO}\tau_{HC}[NO_2]F_J}{k_{NO+O_3}} \times \sqrt{\frac{\alpha}{k_{HO_2+HO_2}[O_3]}} \times J(O^1D)$$

$$J(NO_2) = F_J\sqrt{J(O^1D)}$$

Notably, the observed strong linear relation of OH radicals to J(O$^1$D) is a direct consequence of the efficient recycling of OH by way of the reaction HO$_2$+NO, implying that the empirical model to calculate OH (Eq. 2) may apply to the polluted area—for example, the SORPES station in our study.

For Eq. (2), The dependence of OH on reactants such as NO$_x$, hydrocarbons, ozone or H$_2$O is condensed into the single pre-exponential coefficient, **a**; The exponent **b** reflects the combined effects of all photolytic processes—for example, the photolysis of ozone, NO$_2$, HONO, H$_2$O$_2$ and HCHO. Each of these processes generates OH either directly or via production of and recycling from HO$_2$, and all are highly correlated but not necessarily in a linear manner; the parameter **c** includes all processes that are light-independent—for example, the reactions of ozone with alkenes.

In addition, other dependencies of OH (for example, on NO$_x$ and VOCs concentrations) can result in the variability of the correlation plot of OH with J(O$^1$D), i.e. the variability of the coefficients **a**, **b** and **c** of Eq. (2), but the diversity is not as big as we thought based on the measurements. As listed in Table S1, the value of coefficient **b** is indeed mainly distributed around 1; the value of coefficient **a** does not show a great difference in different seasons or regions; It seems that the value of coefficient **c** in polluted areas is higher than that in clean areas, and the value of coefficient **c** in summer is higher than that in winter, probably due to variability of the strength of nighttime ozone or NO$_3$ radicals reacting with biogenic or anthropogenic VOCs.

To further show how the OH–J(O$^1$D) relation results in a linear dependence, a sensitivity analysis is complemented, showed in Supplementary Table 5 in Rohrer and Berresheim (2006). Except for the combined photolysis frequencies, all parameters (levels of NO$_x$, CO, ozone, VOCs and so on) have very much attenuated influence on OH. Although there are still uncertainty for this conclusion resulting from the simplified reaction scheme for the chemical regime, we can find some evidences from the OH observations in China. As showed in the following figure, the functional dependence of OH$_{Jnorm}$ on NO$_x$ for the observed dataset shows a relatively flat curve, which cannot be predicted by models.

[Figure]

*The solid lines give the averaged values of the measured (red) and modeled (blue) $OH_{Jnorm}$ over equally spaced ln([NOx]/ppb) intervals at the Pearl River Delta (PRD) (Lu et al., 2012)and in Beijing (Lu et al., 2013).For inspection of the OH dependence on NOx, removing the strong influence of j($O^1D$) by normalization as shown in:* $[OH_{Jnorm}] = \dfrac{[OH]}{J(O^1D)}\overline{J(O^1D)}$

Focusing on the OH campaigns in China (Table S1), we can find the approximately equal slope of the OH–J($O^1$D) relation (i.e. the coefficient **a**), on various spatial (at the PRD and in Beijing) and temporal scales (in summer and winter in Beijing). This may can caused by two reasons: One is that the comprehensive impact of reactants (e.g. $NO_x$ and VOCs) on OH cannot compete with that of UV light to OH. Another is that the chemical environments of OH are similar in polluted area of China, and have no great difference in different seasons. The not significant diversity of the coefficient **a** has been proved partly from observations and models in Rohrer and Berresheim (2006). The OH radicals seems to be regulated in such a way that its relation to the driving force—solar radiation—is stabilized in a characteristic functional dependence.

[Figure]

*Correlation of observed (obs) hydroxyl-radical concentrations with simultaneously observed ozone photolysis frequencies ($J(O^1D)$) for the campaigns at the Pearl River Delta (PRD) and in Beijing. The most striking feature is the equal slope of this relation ($4.0 \times 10^{11}$ $cm^{-3}$ s, uncertainty 20%) for both locations, which are 2,000 km apart (Rohrer et al., 2014).*

Moreover, we estimate the uncertainty of OH radicals caused by Eq. (2) (Figure S1(a)) and the influence of the uncertainty of calculated OH on the strength of missing daytime source ($P_{unknown}$). In the case of low concentration of NO and high value of $J(O^1D)$, the error of $P_{unknown}$ from OH radicals will be restrained significantly (Figure S1(d)).

In summary, the empirical model to calculate the concentrations of OH radicals can be reasonable to a certain extent and the error of derived OH radicals has been assessed as not subverting the relative conclusions in this study (Fig. S1(a) and Fig. S1(d)).

[Figure]

**Fig. S1.** (a) The average concentration and standard deviation of OH radicals derived from Eq. (2) by changing the values of the coefficients **a**, **b**, **c**; (b) the relationship of the calculated J(HONO) and J(O$^1$D); (c) the relationship of observed HONO and NO based on diurnal variations (10:00-14:00 LT); (d) the standard deviation of the strength of missing daytime source (Std of P$_{unknown}$) under various values of J(O$^1$D) and NO, and the solid isolines show the ratio of standard deviation to mean value of P$_{unknown}$. The colored and shaped markers in (c) and (d) represent the situations in different seasons.

**2. Response to why the second term (- $k_{NO+OH}$[NO][OH]) should be included on the right side of Eq. 3**

During the daytime, the photolysis of HONO is a reversible reaction, and HONO can be recycled by the homogeneous gas phase reaction of NO with OH. For a given concentration of HONO, when only considering the forward and backward reaction, not all HONO contributes to OH radicals eventually. So we calculate the net OH production rate of HONO by Eq. 3.

$$HONO \xleftrightarrow{UV} NO + OH$$

**3. Response to the meaning of $P_{unknown}$ in Eq. (10)**

Eq. (10) is a deformation of Eq. (9) without minor terms ($T_v$ and $T_h$):

$$\frac{dHONO}{dt} = (P_{NO+OH} + P_{emis} + P_{unknown}) - (L_{phot} + L_{HONO+OH} + L_{dep}) + T_v + T_h$$

$$P_{NO+OH} = k_{NO+OH}[NO][OH]$$

$$P_{emis} = \frac{0.79\% \times \Delta NO_x}{\Delta t}$$

$$P_{NO+OH} = k_{NO+OH}[NO][OH]$$

$$L_{phot} = J(HONO)[HONO]$$

$$L_{HONO+OH} = k_{HONO+OH}[HONO][OH]$$

$$L_{dep} = \frac{\nu_{HONO}}{H}[HONO]$$

So we can get the strength of missing source of HONO at daytime:

$$P_{unknown} = J(HONO)[HONO] + k_{HONO+OH}[HONO][OH] + \frac{\nu_{HONO}}{H}[HONO]$$
$$+ \frac{\Delta HONO}{\Delta t} - k_{NO+OH}[NO][OH] - \frac{0.79\% \times \Delta NO_x}{\Delta t}$$

**Sincerely,**
**Wei Nie**

**Reference**

Lu, K. D., Rohrer, F., Holland, F., Fuchs, H., Bohn, B., Brauers, T., Chang, C. C., Haseler, R., Hu, M., Kita, K., Kondo, Y., Li, X., Lou, S. R., Nehr, S., Shao, M., Zeng, L. M., Wahner, A., Zhang, Y. H., and Hofzumahaus, A.: Observation and modelling of OH and HO2 concentrations in the Pearl River Delta 2006: a missing OH source in a VOC rich atmosphere, Atmospheric Chemistry and Physics, 12, 1541-1569, 10.5194/acp-12-1541-2012, 2012.

Lu, K. D., Hofzumahaus, A., Holland, F., Bohn, B., Brauers, T., Fuchs, H., Hu, M., Haseler, R., Kita, K., Kondo, Y., Li, X., Lou, S. R., Oebel, A., Shao, M., Zeng, L. M., Wahner, A., Zhu, T., Zhang, Y. H., and Rohrer, F.: Missing OH source in a suburban environment near Beijing: observed and modelled OH and HO2 concentrations in summer 2006, Atmospheric Chemistry and Physics, 13, 1057-1080, 10.5194/acp-13-1057-2013, 2013.

Rohrer, F., and Berresheim, H.: Strong correlation between levels of tropospheric hydroxyl radicals and solar ultraviolet radiation, Nature, 442, 184, 10.1038/nature04924 https://www.nature.com/articles/nature04924#supplementary-information, 2006.

Rohrer, F., Lu, K., Hofzumahaus, A., Bohn, B., Brauers, T., Chang, C.-C., Fuchs, H., Häseler, R., Holland, F., Hu, M., Kita, K., Kondo, Y., Li, X., Lou, S., Oebel, A., Shao, M., Zeng, L., Zhu, T., Zhang, Y., and Wahner, A.: Maximum efficiency in the hydroxyl-radical-based self-cleansing of the troposphere, Nature Geoscience, 7, 559-563, 10.1038/ngeo2199, 2014.

Tan, Z., Rohrer, F., Lu, K., Ma, X., Bohn, B., Broch, S., Dong, H., Fuchs, H., Gkatzelis, G. I., Hofzumahaus, A., Holland, F., Li, X., Liu, Y., Liu, Y., Novelli, A., Shao, M., Wang, H., Wu, Y., Zeng, L., Hu, M., Kiendler-Scharr, A., Wahner, A., and Zhang, Y.: Wintertime photochemistry in Beijing: observations of ROx radical concentrations in the North China Plain during the BEST-ONE campaign, Atmospheric Chemistry and Physics, 18, 12391-12411, 10.5194/acp-18-12391-2018, 2018.

---

## Author Response (AR3)

Dear editor,

Thanks for your comment and suggestion. We agree that the OH production rate of HONO should just contains the term of [HONO]J(HONO) from the point of view of chemical kinetics. When we consider a net OH production rate for the reversible reaction between HONO and OH radicals, the term of $k_{HONO+OH}$[HONO][OH] should not be taken into account by definition. And for the term of $k_{NO+OH}$[NO][OH], since the ambient NO and OH are not all from the photolysis of HONO, so it's actually difficult to figure out the net OH production rate for the HONO-NO-OH reaction system in our study. We have corrected Equation 3 and related calculation results in the revised manuscript.

What we want to point out here is that, the $P_{OH}$(HONO) derived from the new Eq. (3) cannot stand for the contribution of HONO to OH. Assuming an extreme case that all of HONO come from the reaction of NO with OH. The photolysis of HONO in this case only regenerates OH radicals, cannot lead to a net increase to the concentrations of OH radicals.

*Line 281-286: As discussed in Su et al. (2008) and Li et al. (2014), HONO produced by the reaction of NO with OH (R3) is actually a temporary reservoir of OH radicals. The photolysis of HONO from this pathway only regenerates OH radicals, cannot contribute to the concentrations of OH radicals. So it is inappropriate to estimate the primary OH production from HONO based on $P_{OH}$(HONO) derived from Eq. (3).*

Sincerely,
Wei Nie

**Reference**

Li, X., Rohrer, F., Hofzumahaus, A., Brauers, T., Haseler, R., Bohn, B., Broch, S., Fuchs, H., Gomm, S., Holland, F., Jager, J., Kaiser, J., Keutsch, F. N., Lohse, I., Lu, K. D., Tillmann, R., Wegener, R., Wolfe, G. M., Mentel, T. F., Kiendler-Scharr, A., and Wahner, A.: Missing Gas-Phase Source of HONO Inferred from Zeppelin Measurements in the Troposphere, Science, 344, 292-296, 10.1126/science.1248999, 2014.
Su, H., Cheng, Y. F., Shao, M., Gao, D. F., Yu, Z. Y., Zeng, L. M., Slanina, J., Zhang, Y. H., and Wiedensohler, A.: Nitrous acid (HONO) and its daytime sources at a rural site during the 2004 PRIDE-PRD experiment in China, Journal of Geophysical Research, 113, 10.1029/2007jd009060, 2008.